# Hazard Interactions and Interaction Networks (Cascades) within Multi-Hazard Methodologies

Joel. C. Gill[1], Bruce. D. Malamud[1]

[1]Department of Geography, King's College London, London, WC2R 2LS, UK

*Correspondence to*: Joel. C. Gill (joel.gill@kcl.ac.uk)

**Abstract.** This paper combines research and commentary to reinforce the importance of integrating hazard interactions and interaction networks (cascades) into multi-hazard methodologies. We present a synthesis of the differences between 'multi-layer single hazard' approaches and 'multi-hazard' approaches that integrate such interactions. This synthesis suggests that ignoring interactions between important environmental and anthropogenic processes could distort management priorities,

increase vulnerability to other spatially relevant hazards or underestimate disaster risk. In this paper we proceed to present an enhanced multi-hazard framework, through the following steps: (i) describe and define three groups (natural hazards, anthropogenic processes and technological hazards/disasters) as relevant components of a multi-hazard environment; (ii) outline three types of interaction relationship (triggering, increased probability, and catalysis/impedance); and (iii) assess the importance of networks of interactions (cascades) through case-study examples (based on literature, field observations and

semi-structured interviews). We further propose two visualisation frameworks to represent these networks of interactions: hazard interaction matrices and hazard/process flow diagrams. Our approach reinforces the importance of integrating interactions between different aspects of the Earth system, together with human activity, into enhanced multi-hazard methodologies. Multi-hazard approaches support the holistic assessment of hazard potential, and consequently disaster risk. We conclude by describing three ways by which understanding networks of interactions contributes to the theoretical and

practical understanding of hazards, disaster risk reduction and Earth system management. Understanding interactions and interaction networks helps us to better (i) model the observed reality of disaster events, (ii) constrain potential changes in physical and social vulnerability between successive hazards, and (iii) prioritise resource allocation for mitigation and disaster risk reduction.

## 1 Introduction

In this article we present both research and commentary to support the integration of hazard interactions and their networks (cascades) into multi-hazard methodologies. Building on the work of others (Delmonaco et al., 2007 Kappes et al., 2010; Kappes et al., 2012; Marzocchi et al., 2012; Gill and Malamud, 2014) we advocate for a multi-hazard approach that goes beyond the simple overlay of multiple single hazards, to an approach that also encompasses interactions between these hazards. We present here an enhanced framework for considering such interactions and integrating these into multi-hazard

methodologies, supporting efforts to improve management of those aspects of the Earth system that are relevant to disaster risk reduction. Examples from primary research and published literature, together with commentary about multi-hazard approaches, are included throughout.

Following this introduction, **Sect. 2** examines the differences between single hazard, multi-layer single hazard, and full multi-hazard risk approaches. In **Sect. 3** we define and describe three distinct hazard and process groups (natural hazards, anthropogenic processes, technological hazards/disasters) that can be considered in multi-hazard methodologies. This is followed by **Sect. 4** which discusses and visualises three principal interaction relationships between these hazards and processes (triggering relationships, increased probability relationships, catalysis/impedance relationships), with a detailed description of their differences and examples of each. Then in **Sect. 5** we discuss how individual interactions can join together to form networks of hazard interactions (cascades), using four case studies (one from Nepal and three from Guatemala) and two theoretical examples, to consider different features of interaction networks and how these can be visualised using hazard interaction matrices and hazard/process flow diagrams. We also commentate on the benefits of assessing networks of hazard interactions to support disaster risk reduction. Conclusions are outlined in **Sect. 6**.

## 2 Multi-Hazard Risk Assessments

### 2.1 Single vs. Multi-Hazard

Single hazard approaches to assessing hazard potential, in which hazards are treated as isolated and independent phenomena, are commonplace. Their prevalence, however, can distort management priorities, increase vulnerability to other spatially relevant hazards or underestimate risk (Tobin and Montz, 1997; ARMONIA, 2007; Kappes et al., 2010; Budimir et al., 2014; Mignan et al., 2014). If a community is susceptible to more than one hazard, management decisions will benefit by reflecting the differential hazard potential and risk from each of these, and not just focus on them as individual entities. Focusing on a small portion of the whole Earth system, rather than the dynamics of its entirety, may result in decisions being made that increase people's vulnerability to other, ignored hazards. The development of enhanced 'multi-hazard' risk approaches (integrating all aspects of hazard interactions together with vulnerability and exposure) could offer a way by which the disaster risk reduction community can address these problems.

Multi-hazard approaches are widely encouraged in key government and intergovernmental initiatives and agencies, but rarely defined. For example, the Hyogo Framework for Action (2005–2015) called for *"an integrated multi-hazard approach to disaster risk reduction"* (UNISDR, 2005, p. 4). The Sendai Framework for Disaster Risk Reduction (2015–2030) states that *"disaster risk reduction needs to be multi-hazard"* (UNISDR, 2015, p. 10). Despite the emphasis on multi-hazard approaches within these international agreements, both the Hyogo and Sendai Frameworks do not define what a multi-hazard approach involves. At the time of writing, the term multi-hazard also does not appear in the most recent descriptions of terminology

published by UNISDR (2009). Further examples of multi-hazard approaches being advocated for, but not clearly defined, can be found in United Nations (2002) and Government Office for Science (2012).

The term 'multi-hazard' may appear to be unambiguous to some and not require a definition. It is, however, a term that is frequently used in different contexts by different members of the natural hazards and disaster risk reduction community. It has
been used to describe the independent analysis of multiple different hazards (e.g., landslides, earthquakes, pyroclastic density currents, tephra fall, flooding) relevant to a given area (e.g., Granger et al., 1999; Perry and Lindell, 2008). It has also been used when referring to the identification of areas of spatial overlap, by superimposing hazard layers (e.g., Dilley et al., 2005; Shi et al., 2015). These can be better thought of as 'multi-layer single hazard' approaches (Gill and Malamud, 2014), where an 'all-hazards-at-a-place' framework (Hewitt and Burton, 1971) seeks to understand the discrete risks due to multiple natural
hazards.

The identification of all possible and spatially relevant hazards is an important feature of a full multi-hazard assessment, but we believe should not be the sole defining characteristic of such an approach. Multi-hazard assessments may also recognise the non-independence of natural hazards (Kappes et al., 2010), noting that significant interactions exist between individual natural hazards. In a previous study (Gill and Malamud, 2014) we took 21 different natural hazards and identified 90 possible
interactions between the 441 ($21 \times 21$) combinations. Here, we will further consider (**Sect. 3–4**) interactions that may also exist between natural hazards, anthropogenic processes (human activity) and the built environment. We will also consider (**Sect. 5**) interactions that can occur successively to form networks of hazard interactions, also referred to as hazard cascades or chains (e.g., Xu et al., 2014; Choine et al., 2015).

We now highlight five possible types of hazard interactions that may occur if an inhabited location is susceptible to multiple
hazards, using as exemplars four natural hazards (tropical storms, floods, landslides and volcanic eruptions):

    i.    *Natural hazards triggering other natural hazards*: For example, a tropical storm (primary natural hazard) may trigger secondary natural hazards, such as flooding, landslides or lahars if there has been a recent volcanic eruption of tephra.

    ii.    *Human activities triggering natural hazards:* For example, road construction may destabilise a slope and trigger a landslide.

iii.    *Human activities exacerbating natural hazard triggering*: For example, deforestation may exacerbate the triggering of landslides or floods (secondary natural hazards) during a tropical storm (primary natural hazard).

    iv.    *Networks of hazard interactions (cascades):* For example, a tropical storm (primary natural hazard) may trigger hundreds of landslides (second natural hazard), some of which may dam rivers and exacerbate flooding. This in turn could cause slope erosion and trigger further landslides.

v.    *The concurrence of two (or more) hazard events*: For example, the spatial and temporal overlap of a volcanic eruption and tropical storm event may result in flooding of a greater severity than would have occurred otherwise, due to volcanic ash blocking drainage systems.

The above five interaction types, based on just four natural hazard exemplars, are taken from a much broader range of possible hazard interactions and their networks. Even with these limited examples, they demonstrate the limitations of assuming independence of single hazards within a multi-layer single hazard approach.

Multi-hazard methodologies, therefore, should ideally evaluate all identified individual hazards relevant to a defined spatial area and characterize all possible interactions between these identified hazards. **Figure 1**, from Gill and Malamud (2014) shows four distinct factors required to transition from a multi-layer single hazard assessment to a detailed, full multi-hazard risk assessment (which includes hazard interactions, vulnerability and exposure). In addition to identifying all hazards and their interactions, this working framework also proposes an assessment of concurrent hazards (such as a tropical storm and volcanic eruption coinciding spatially and temporally), and the recognition that vulnerability is dynamic (which we discuss more in **Sect. 5.3**).

Many current hazard assessments that are labelled as 'multi-hazard' do not consider all the factors given in **Figure 1**, in either a qualitative or quantitative manner. This may be a consequence of limited existing methodologies to assess each of the steps proposed in **Figure 1** of a multi-hazard approach. Those methodologies that do exist are sometimes complex, requiring significant amounts of data. Some accessible methodologies to allow the comparison of natural hazards, however, can be found within the literature (e.g., Granger et al., 1999; Van Westen et al., 2002; Greiving et al., 2006; Grunthal et al., 2006; Marzocchi et al., 2009). Methodologies to identify and visualise potential natural hazard interactions also exist (e.g., Tarvainen et al., 2006; Han et al., 2007; De Pippo et al., 2008; Kappes et al., 2010; van Westen et al., 2014, Gill and Malamud, 2014, Liu et al., 2016), including a progression towards more quantitative approaches (e.g., Neri et al., 2013; Marzocchi et al., 2012). In this paper, we will consider multi-hazard risk frameworks, with a focus on the hazard interaction component of the risk framework (and not so much of a focus on vulnerability and exposure).

## 2.2 From Global to Local Multi-Hazard Approaches

The hazard interactions literature outlined in **Sect. 2.1** includes studies for different spatial extents, including global (e.g., Gill and Malamud, 2014), continental (e.g., Tarvainen et al., 2006) and local or sub-national (e.g., De Pippo et al., 2008). The scale of interest for a particular multi-hazard approach determines how interactions are characterised. Approaches may be based on an examination of an individual event (e.g., a given earthquake triggering landslides in a given region), or draw on a large population of individual events to infer the probabilistic behaviour of a relationship (e.g., considering many earthquake triggered landslide events over different regions, and from this the dependence of the number of landslides triggered based on the earthquake magnitude). The latter approach is used to consider in general how one hazard will influence another. Both approaches are beneficial in different contexts. For example, a probabilistic viewpoint is likely to support the characterisation of possible interactions in a general, globally relevant way, as we often consider them in our paper. When adapting global, multi-hazard approaches for use in regional and local contexts, a different population of individual events is required to infer the probabilistic behaviour of a relationship specific to that context. In many regions, although the database of events is likely

to better reflect site specific conditions, it may be small, consisting of just a couple (sometimes zero) individual events, depending on the period of time considered.

Another possible contrast between globally-relevant multi-hazard approaches and location specific, multi-hazard approaches is the forecasting time window (Marzocchi et al., 2012) or temporal resolution (Kappes et al., 2012). In globally-relevant approaches that draw upon many individual events, generalisations across forecasting time windows (both short- and long-term time windows) are used to construct the multi-hazard framework, with the inclusion of interactions relevant at all temporal resolutions. When developing location specific multi-hazard assessments, clear temporal limits should be established (Selva, 2013), depending on the purpose of the multi-hazard approach. When constructing location-specific assessments of hazard potential, Marzocchi et al. (2012) proposes that the modelling of hazard interactions is more necessary in the short term (e.g., hours to days) than the long term (e.g., many decades to centuries). They argue that in the short term, the occurrence of a primary hazard (e.g., storms) can significantly modify the probability of secondary hazards (e.g., floods, landslides), compared to the long-term, where primary hazards (e.g., earthquakes, landslides) are already considered in the long-term assessment of the secondary hazard (e.g., tsunamis). In other words, they discuss that in a long-term perspective (e.g., the tsunami hazard over the next 50 years), databases already contain information of the fact that most tsunamis are triggered by earthquakes, and there is no need to make additional calculations to calculate the long-term tsunami hazard. It is therefore less necessary in the long term (compared to short term) to model possible interactions as databases of past single hazard occurrences already reflect the triggered nature of these hazards. In the long term, however, it is important to consider the temporal proximity of successively occurring hazards (e.g., earthquake → tsunami) in order to better evaluate possible risks and losses. We will further explore short-term and long-term time windows in relation to natural hazard interactions (primary hazard triggering a secondary hazard) in **Section 4.1**. The importance of interactions in both short and long-term contexts can aid the understanding of natural hazards, hazard education, communication and disaster risk reduction.

As further multi-hazard approaches are developed, and integrated into research and practice, we believe that it is important to recognise that (i) natural hazards do not operate in isolation, (ii) the characteristics of a framework at global spatial scales may differ from more context/location-specific scales, and (iii) enhanced multi-hazard approaches would also likely benefit from considering how human activity can influence the triggering of hazards and initiation of networks of hazard interactions. We now proceed to define and describe three principal groups of hazards and processes that enhanced multi-hazard frameworks may consider including.

## 3 Hazard and Process Groups

Here we discuss the characterisation of hazard potential for an applied multi-hazard approach that includes an assessment of at least three distinct groups: (i) natural hazards, (ii) anthropogenic processes and (iii) technological hazards/disasters. All of

these can be considered to be processes and/or phenomena with the potential to have negative impacts on society. In the context of this article, these terms are defined as follows:

i. *Natural hazards.* A natural process or phenomenon that may have negative impacts on society (UNISDR, 2009). Examples include earthquakes, volcanic eruptions, landslides, floods, subsidence, tropical storms and wildfires.

ii. *Anthropogenic processes.* Intentional human activity that is non-malicious, but that may have a negative impact on society through the triggering or catalysing of other hazardous processes. The word process here (and used in many other places in the text) is taken to mean "*a continuous and regular action or succession of actions occurring or performed in a definite manner, and having a particular result or outcome; a sustained operation or series of operations*" (OED, 2015). Examples include groundwater abstraction, vegetation removal, quarrying and surface mining, urbanisation and subsurface construction (tunnelling).

iii. *Technological hazards/disasters.* The unintentional, non-malicious or negligent failure of technology or industry. Examples include structural collapse, nuclear reactor failure, urban fires, chemical pollution and dam collapse.

A more detailed list of examples for each of these three groups (natural hazards, anthropogenic processes, technological hazards/disasters), based on the definitions set out above, are given in **Table 1**. We now discuss in more detail (**Sect. 3.1–3.3**) each of these three groups, particularly potential overlap between the words 'anthropogenic process' and 'technological hazard' with additional brief comments in **Sect. 3.4**.

### 3.1 Natural Hazards

The meaning of the phrase natural hazards, considered both individually and as a group of processes is reasonably well understood (e.g., Alexander, 1993; Smith, 2013). The broad definition of a natural hazard, as set out by UNISDR (2009), is well accepted and encompasses those natural processes that are widely considered to potentially have a negative impact on society and the natural environment. Differences may exist in the level of organisation, or the resolution of classification, used to describe each single hazard. For example, in their National Risk Register, the UK Cabinet Office (2013) divides floods into coastal flooding and in-land flooding. Differences may also exist in how single hazards are clustered. For example, landslides may be clustered with other single hazards within one or more of the following broader hazard types: geophysical, geomorphological, hydrological, and/or hydro-meteorological. These differences in resolution of classification and clustering are normally due to different purposes and characteristics of interest to a specific project, rather than any significant differences of understanding in the process.

### 3.2 Anthropogenic Processes

Anthropogenic processes are less well defined and characterised as a group, compared to the group labelled 'natural hazards'. There are numerous references to individual human activities exacerbating or triggering particular natural hazards in the

literature. For example, Owen et al. (2008) refers to the role of road construction in exacerbating landslide initiation during the 8 October 2005 Kashmir earthquake; Glade (2003) refers to the role of land cover changes in the triggering of landslides during rainstorms in New Zealand; and Knapen et al. (2006) refers to the role of vegetation removal in triggering landslides in Uganda. Induced seismicity is a further example of an anthropogenic process triggering a natural hazard. Anthropogenic

processes believed to induce seismicity include reservoir construction (Simpson, 1976), groundwater abstraction (Gonzáles et al., 2012), and wastewater injection (Ellsworth, 2013; Hough and Page, 2015). Each of these examples involves an intentional, non-malicious human activity that has the potential to have a negative impact on society through the triggering or catalysing of hazards. UNISDR (2009) defines the occurrence of specific natural hazards arising from overexploited or degraded natural resources as 'socio-natural' hazards. By definition, these are generated by the interaction of anthropogenic processes with the

natural environment. The inclusion of anthropogenic processes within multi-hazard approaches is therefore important and justified. They are very relevant to the modelling of Earth system dynamics and hazardous environments.

### 3.3 Technological Hazards/Disasters

Although often referred to in the context of disaster studies (e.g., Fleischhauer, 2006; Tarvainen et al., 2006; Bickerstaff and Simmons, 2009), technological hazards/disasters are also less well defined and characterised than the group 'natural hazards'.

Some definitions or descriptions of technological hazards and disasters do exist (e.g., Kasperson and Pijawka, 1985; Gunn, 1990; UNISDR, 2009), but these often lack clarity, or conflict with one another. For example, some definitions include intentional anthropogenic activities within their definition of technological hazards/disasters. Gunn (1990) refers to technological disasters as being human-initiated consequences of breakdown, technical fault, errors, or involuntary and voluntary human acts that have negative consequences. The latter (voluntary human acts) includes those examples that we

have defined in **Sect. 3.2** as anthropogenic processes. Subsurface mining, for example, is a voluntary human act that can result in environmental damage, such as subsidence. This subsidence can vary in intensity from slight to severe (Bell et al., 2000).

The UNISDR (2009) definition of technological hazards also states that hazards originate from technological or industrial conditions, including human activities that may cause environmental damage, health impacts, economic disruption and other negative consequences. This could include human activities such as subsurface mining, groundwater abstraction and vegetation

removal. Therefore, the UNISDR (2009) definition of technological hazards also appears to include examples that we have categorised as anthropogenic processes.

Other authors make a clearer distinction between anthropogenic processes and technological hazards. For example, Kasperson and Pijawka (1985) outline three categories of technological hazards:

   i.   *Routine hazard events of technology*, where there is exposure to underlying chronic hazardous activity over an

30        extensive time period. These can normally be managed by established procedures.

   ii.  *Technology failures*, resulting in the need for an emergency response.

iii. *Technological disasters*, resulting in significant loss of life or injury, social disruption or relocation.

The latter two (technology failures, technological disasters) are distinguished based on the scale of impact, with technological failures able to evolve into technological disasters if losses are sufficiently large. Although included within the broad category of technological hazards in Kasperson and Pijawka (1985), there is significant overlap between their definition of *routine*
*hazard events of technology* and our definition of anthropogenic processes, outlined in **Sect. 3.2**. For example, in **Table 1** we note surface mining to be an anthropogenic process. This classification is based on our definition of anthropogenic processes being intentional human activities that are non-malicious but may have a negative impact on society through the triggering or catalysing of hazardous processes (**Sect. 3**). Surface mining can also be considered to be a *routine hazard event of technology* as defined by Kasperson and Pijawka (1985), in that the mining is a technological process where there is exposure to underlying
chronic hazardous activity, which can be managed by established procedures.

Whereas technological failures and disasters are generally *unintentional* (i.e., not a result of a conscious choice or a desired process), anthropogenic processes are generally *intentional*, and are a result of conscious decisions that may subsequently result in negative consequences. Although such consequences can often be managed using established procedures, anthropogenic processes sometimes still result in the triggering or catalysing of a natural hazard. In the context of this article,
therefore, technological hazards are taken to be unintentional, non-malicious or negligent failures of technology or industry.

**3.4 Additional Hazards or Processes**

In **Sect. 3.1–3.3** it was noted that both anthropogenic processes and technological hazards/disasters are non-malicious; the negative consequences are not the desired outcome. Events that are malicious or deliberately destructive (e.g., terrorism, arson, aspects of warfare and criminal activity) are not included within either 'anthropogenic processes' or 'technological
hazards/disasters', but may trigger the occurrence of other hazards or processes. For example, the deliberate, and malicious detonation of a bomb close to a dam (this is not an anthropogenic process, as it is malicious) may trigger the dam to collapse (technological hazard), resulting in substantial flooding (natural hazard).

In the context of the rest of this article we focus on interaction relationships between the three groups just discussed— natural hazards, anthropogenic processes and technological hazards/disasters—and the development of possible networks of hazard
interactions (cascades).

**4 Interaction Relationships**

**4.1 Classifying Interaction Types**

Multiple interactions exist between the hazard and process examples outlined in the three groups (natural hazards, anthropogenic processes, technological hazards/disasters) discussed above. Kappes et al. (2012) notes a wide variety of terms
used to describe such interactions (e.g., interrelationships, interconnections, coupled events) and specific sets of interacting

hazards (e.g., coinciding hazards, triggering effects). Here we continue to use the term 'interactions' to describe the possible inter- and intra-relationships for hazards and processes. We note that the term 'interaction' communicates the potential for unidirectional and bidirectional relationships. In unidirectional relationships first the 'primary' hazard occurs and then the 'secondary' hazard. An example is a tropical storm triggering a flood. In this case the flood may trigger further hazards (e.g.,

ground collapse, ground heave), but there is no feedback from the flood back to the tropical storm. In bidirectional relationships, feedback mechanisms may occur where a primary hazard triggers a secondary hazard which exacerbates the primary hazard, therefore triggering further episodes of the secondary hazard. An example of this would be a landslide blocking a river, resulting in a flood, but then the water upstream of the blockage interacting with the original landslide, breaking it down, and the water potentially triggering further landslides.

We use the term 'interaction', therefore, to refer to the unidirectional and bidirectional effect(s) between one hazard/process and another hazard/process, and note examples of three distinct types of interaction relationships:

    i.    *Triggering relationships* (e.g., lightning triggering a wildfire; groundwater abstraction triggering regional subsidence; a flood triggering a landslide which then triggers a further flood).

    ii.    *Increased probability relationships* (e.g., a wildfire increasing the probability of landslides; regional subsidence

15             increasing the probability of flooding).

    iii.    *Catalysis/impedance relationships* (e.g., urbanisation catalysing storm triggered flooding; storms impeding urban fire triggered structural collapse).

While we distinguish *triggering* relationships and *increased probability* relationships as two different types of interactions, we acknowledge that there are similarities between them as they both represent a change in probability of a secondary hazard

(e.g., landslide), given a primary hazard (e.g., earthquake). We would suggest that these two interaction types can be characterised by two end-members, with a continuum between them. A *triggering* relationship can be characterised as having a probability associated with a threshold being reached or passed. An *increased probability* relationship is characterised by a probability associated with a change in environmental parameters that moves towards, but does not reach a particular threshold. Although there are similarities, we would suggest that it is beneficial to consider both triggering and increased probability

relationships as separate interaction types. We propose two ways by which one can differentiate between a hazard/process triggering another hazard/process, and a hazard/process increasing the probability of another hazard/process:

    i.    *Direct vs. indirect sequence of events between the primary and secondary hazard*. In some cases, it is possible to differentiate between triggering and increased probability relationships by considering direct vs. indirect sequences of events between the primary and secondary hazards/processes. An example of a (roughly) direct sequence is the

30        addition of water to geological material on a hillslope, which can directly trigger landslides (heavy rain → landslides). In contrast, an example (roughly) of an indirect sequence is the influence of ground subsidence on flooding. Subsidence in itself may not trigger a flood; however, it could make flooding more likely to occur in the event of a

river spilling over its banks. Direct sequences tend to be triggering relationships, whereas indirect sequences tend to be increased probability relationships.

ii. *Temporal sequence.* It is also possible to differentiate between some triggering and increased probability relationships by considering the timing of the sequence of events, and taking both forward looking and retrospective views. Take the following time sequence (arbitrary units and lengths of time for the windows):

**Time Window 1** | *[Primary Hazard Window]* | **Time Window 2A** | **Time Window 2B**

As an illustrative example, we will take an earthquake as the primary hazard, and a triggered landslide population event as the secondary hazard, and discuss perspectives from before the primary hazard occurs (**Time Windows 1)** and two time periods after the primary hazard occurs (**Time Windows 2A** and **2B**):

- **While in Time Window 1, consider what may happen in Time Windows 2A and 2B** [forward looking]. Prior to an earthquake [primary hazard] occurring [**Time Window 1**], and based on past historical knowledge of the region (e.g., a 50 year historical catalogue of past earthquakes), it can be stated that there is a given probability of an earthquake occurring and that given an earthquake, landslides [secondary hazard] may occur (i.e., they are triggered by the earthquake) in the time period of minutes to days [**Time Window 2A**] after the earthquake. Furthermore, while in **Time Window 1**, we can state that more landslides may occur (i.e., an increased probability) much later after an earthquake event (months to years) [**Time Window 2B**] due to changes in the parameters governing the stability of the slope (Havenith, 2014). We cannot know if landslides [secondary hazard] will be directly triggered by the earthquake [primary hazard] until after the earthquake has occurred (i.e., until the end of **Time Window 2A**), but we can postulate that the earthquake might trigger landslides.

- **While at the end of Time Window 2A, consider what has occurred in Time Window 2A** [retrospective] **and what may happen in Time Window 2B** [forward looking]. Looking retrospectively at **Time Window 2A**, the period just after the earthquake [primary hazard] has occurred, we have identified any landslides [secondary hazard] that were triggered by the earthquake. We can also look forward to **Time Window 2B** and state that there is now an increased probability of landslides due to changes in the parameters governing the stability of slopes in the region.

- **While at the end of Time Window 2B, consider what has occurred in Time Windows 2A and 2B** [retrospective]. At this position in time, we can retrospectively assess what landslides have been triggered by the earthquake, either directly triggered in the minutes to days after the earthquake event (**Time Window 2A**), or earthquake induced changes to the landscape which result in broader changes to landslide susceptibility over longer time periods (**Time Window 2B**).

When generalising across these three time windows (**1, 2A, 2B**), recognising that an earthquake [primary hazard] can both trigger and increase the likelihood of landslides [secondary hazard] occurring in [**Time Windows 2A and 2B**] can be a useful concept, particularly for decision making at an operational level.

In summary, while causal triggering relationships can only be 'known' retrospectively, there is still a good justification for distinguishing between triggering and increased probability relationships when using forward-looking approaches. For any given window of time after a primary hazard, those interested in hazard interactions (e.g., scientists, hazard managers) may want to know what the likelihood is of landslides occurring (being triggered), but also whether there is a change in the likelihood of landslides beyond this window of time (increased probability). Although in some cases, attribution, or identifying a causal relationship between a specific primary hazard (e.g., a given earthquake) and a specific secondary hazard (e.g., a given tsunami) is clear, other times attribution is not so clear (e.g., the increase of probability of landslides as a result of a wildfire). This challenge of attribution is currently in the forefront of the climate change community, where attempts are made to determine the existence of causal relationships between anthropogenic climate change and specific extreme events (Stott *et al.,* 2013; Shepherd, 2016).

We now discuss each of these three interaction relationships in more detail, giving examples and introducing two visualisations. These interaction relationships are also used in **Sect. 5**, when discussing networks of interactions (cascades).

## 4.2 Triggering Relationships

Triggering relationships are one form of causal relationship, where the occurrence of a primary event can result in secondary events occurring. For example, a tropical storm or hurricane (a primary natural hazard) may trigger many landslides (a secondary natural hazard) due to the rapid increase in ground saturation, such as in the case of Hurricane Mitch in 1998 where heavy rain associated with the hurricane resulted in thousands of landslides being triggered in Guatemala (Bucknam et al., 2001). As noted in **Sect. 4.1**, feedback mechanisms can also exist where a triggered secondary hazard exacerbates the primary hazard and results in further occurrences of the primary and/or secondary hazard being triggered.

Triggering interactions can occur between a diverse range of hazards and processes. Gill and Malamud (2014) considered just natural hazards, and identified 78 possible triggering pairings between 21 natural hazards (the same natural hazards as those given in **Table 1**). The inclusion of both 'anthropogenic processes' and 'technological hazards/disasters' would result in many more triggering relationships than the 78 identified by Gill and Malamud (2014) for natural hazards, as not only would there be triggering relationships within each of the two additional groups ('anthropogenic processes', 'technological hazards/disasters'), but also a significant number would arise between all three groups.

We also highlight that each triggering relationship identified will have different likelihoods associated with it. In any given location, the likelihood will be dependent on site-specific conditions (e.g., geology, hydrology, neotectonics, the extent of human activity). From a probabilistic viewpoint, generalising across multiple individual events for each triggering relationship,

we can also infer that some triggering relationships are more likely to occur than others. For example, Gill and Malamud (2014) use a nine-point scale to classify the *spatial overlap* and *temporal likelihood* of each of the 78 primary-secondary natural hazard triggering relationships that they identified. An example of a triggering relationship with low *spatial overlap* and low *temporal likelihood* is a landslide triggering a volcanic eruption. An example of a triggering relationship with high *spatial overlap* and high *temporal likelihood* is a storm triggering a landslide.

Of importance in the context of characterising triggering relationships are the spatial and temporal scales of interest. When considering interactions in a specific local/regional setting, different interaction behaviours will occur at different spatial and temporal scales. For example, an anthropogenic process, such as agricultural practice change, could occur at multiple scales. An individual farmer ploughing a new field (approximate spatial scale of $0.1$–$1$ km$^2$, temporal scale of days to weeks) is likely to have a different influence compared to a societal transition from manual to machine-dominated farming (approximate spatial scale of $10^4$–$10^7$ km$^2$, temporal scale of years to centuries). Differences in the scale of interest of agricultural practice change would result in diverse characterisations of the possible triggering relationships. In the context of this paper we are not focusing on a specific local/regional study or at a specific spatial/temporal scale, but rather considering a global overview of interactions, generalising across many spatial and temporal scales.

Another important factor for consideration when characterising triggering relationships, is the relative timing of different stages. For example, some anthropogenic processes may involve multiple stages, including an initial decision-making or survey stage before ground disturbance. In this example, it is possible that a given anthropogenic process may trigger other processes to occur before, simultaneously with, or after any ground disturbance has occurred. Where an associated process is stated to occur 'before' a primary anthropogenic process, it is normally occurring after at least one preliminary stage of the primary anthropogenic process. Associated processes can therefore be considered to be triggered by an occurrence of a primary anthropogenic process. For example, subsurface construction, such as tunnelling, may require drainage and dewatering to take place before the tunnelling commences. The need for drainage or dewatering would be determined during preliminary ground reconnaissance and site investigation.

When considering combinations between the three groups of hazards/processes (natural hazards, anthropogenic processes, technological hazards/disasters), we identify nine possible triggering relationships between these groups and visualise these in **Figure 2**, a hazard/process flow diagram. Triggering relationships are illustrated using block arrows, with the internal arrow fill colour indicating the group of hazards or processes to which the 'trigger source' belongs to. Medium grey is used for natural hazards (labelled **A**), dark grey is used for anthropogenic processes (labelled **B**), and light grey is used for technological hazards/disasters (labelled **C**). We use a prime (**A′**, **B′**, **C′**) to indicate secondary hazards/processes triggered by the same primary hazard or process (**A**, **B**, **C**). Examples of all nine possible interactions are given in a table below **Figure 2**, with codes (i.e., $A_1$–$A_3$, $B_1$–$B_3$, $C_1$–$C_3$) relating to arrow labels derived from the hazard or process type of the 'trigger source' (i.e., **A**, **B**, **C**), and followed by sequential subscript numbering. Numbering in our hazard/process flow diagram starts ($A_1$, $B_1$, $C_1$) with the triggering relationship between the same primary and secondary hazard or process type (e.g., a primary *natural hazard*

triggering a secondary *natural hazard*) and progresses clockwise. These nine possible triggering relationships demonstrate an important set of interaction relationships that could be included within a multi-hazard methodology.

## 4.3 Increased Probability Relationships

Another type of causal relationship can be observed when a primary natural hazard, anthropogenic process or technological hazard increases the probability of another such event occurring. These situations involve a primary hazard or process altering one or more environmental parameters so as to change the temporal proximity or specific characteristics of an individual or population of secondary hazards or processes (Kappes et al., 2010; Gill and Malamud, 2014). Examples relating to specific natural hazards include an earthquake increasing the susceptibility of a slope to landslides, regional subsidence increasing the probability of flooding, or wildfires increasing the probability of ground heave. In Gill and Malamud (2014), we took the 21 different natural hazards identified in **Table 1**, and identified 75 possible relationships where a primary natural hazard could increase the probability of a secondary natural hazard. The inclusion of anthropogenic processes and technological hazards/disasters will also result in many more increased probability relationships.

## 4.4 Catalysing and Impedance Relationships

We have discussed above that one hazard/process may trigger another hazard/process. It is possible that further hazards and processes may cause these triggering relationship pairings to be catalysed or impeded. For example, tropical storms can often trigger floods. This triggering relationship can be catalysed by other specific anthropogenic processes (e.g., vegetation removal, urbanisation), natural hazards (e.g., wildfires) or technological failures (e.g., blocked drainage). Conversely a volcanic eruption can trigger wildfires, but this triggering relationship may be impeded by other specific anthropogenic processes (e.g., deforestation) or natural hazards (e.g., tropical storms).

In addition to the nine triggering interaction relationships previously identified (**Figure 2**), a further 12 possible catalysing and impedance relationships can be considered, which we visualise in **Figure 3**, also a hazard/process flow diagram. In **Figure 3**, we contrast triggering relationships (9 thick block arrows with solid outlines), and catalysing/impedance relationships (12 thin block arrows with dashed outlines). The internal arrow fill colour again indicates the group of hazards or processes to which the catalyst/impeder belongs (medium grey: natural hazards; dark grey: anthropogenic processes; light grey: technological hazards/disasters).

**Figure 3** highlights the range of possible interaction relationships between the three broad groups of hazards/processes, using a hazard/process flow diagram. Within each type of interaction relationship there exist specific interactions that are rare and others that are very common, with a wide spectrum between these two end-members. Location specific conditions influence the likelihood of any given interaction relationship. The likelihood of each catalysing relationship will depend on (i) the likelihood of the primary hazard/process occurring, (ii) the likelihood of the primary hazard/process triggering a secondary hazard, and (iii) the likelihood of a given hazard/process catalysing this interaction pairing. Consider, for example, the

unloading of slopes through road construction catalysing earthquake or storm triggered landslides (thin, dark grey arrow from **B** to **A₁** in **Figure 3**). In **Sect. 3.2** we introduced this example, describing how Owen et al. (2008) had found that road construction catalysed the triggering of landslides during the 8 October 2005 Kashmir earthquake. In regions that are susceptible to landslides, the influence of road construction is well documented (e.g., Montgomery, 1994; Devkota et al., 2013; Brenning et al., 2015). It is a catalysing relationship that is common in many parts of the world. Overall, the differential likelihood of any relationship will depend on the range of location-specific parameters. Only through the careful assessment of all possible single hazards and processes can relevant interactions be identified and assessed.

Examples of some specific catalysing and impeding interaction relationships are presented below. Here we state which hazard or process group (e.g., anthropogenic process) is acting as the catalysing or impeding agent, whether it is a catalysis or impedance relationship, and which triggering relationship identified in **Sect. 4.2** is being catalysed or impeded (e.g., **A₁**, **B₁**, **C₁**, as labelled and described in **Sect. 4.2**). We then give a more specific example.

i. *Anthropogenic processes catalysing triggering relationship* **A₁** (thin dark grey arrow from **B** to **A₁** in **Figure 3**): Example: urbanisation catalyses storm triggered flooding.

ii. *Technological hazards/disasters catalysing triggering relationship* **A₁** (thin light grey arrow from **C** to **A₁** in **Figure 3**): Example: dam collapse catalyses flood triggered landslides.

iii. *Natural hazards catalysing triggering relationship* **B₁** (thin medium grey arrow from **A** to **B₁** in **Figure 3**): Example: floods catalyse urbanisation triggered agricultural practice change.

iv. *Natural hazards impeding triggering relationship* **C₁** (thin medium grey arrow from **A** to **C₁** in **Figure 3**): Example: storm impedes structural collapse triggered urban fires

## 5 Networks of Hazard Interactions (Cascades)

In **Sect. 4**, we discussed three different interaction relationships (triggering, increased probability, catalysing/impedance) between specific natural hazards, anthropogenic processes and technological hazards/disasters. However, in addition to having a paired relationship (e.g., one primary natural hazard triggering a secondary natural hazard) these interactions can be joined together to form a network of hazard and/or process interactions. For simplification of language, we will call these just 'networks of hazard interactions' or 'interaction hazard networks'. Such networks have also been referred to as hazard chains (e.g., Han et al., 2007; Xu et al., 2014), cascades (e.g., Choine et al., 2015; Pescaroli and Alexander, 2015) or multi-hazard networks of interacting hazards (Gill and Malamud, 2014). Networks of hazard interactions may consist of short or long chains of interactions, and may include single or multiple branches.

In **Sect 5.1** we introduce four case study examples of networks of hazard interactions, one example from Nepal and three from Guatemala. In **Sect. 5.2** we illustrate the wide variation in spatial and temporal extent, frequency and impacts of such networks

of hazard interactions, using three of these case studies. In **Sect. 5.3** we then use our hazard interaction matrix and hazard/process flow diagrams to visualise networks of hazard interactions, using two of these case studies and three theoretical examples. Finally, in **Sect 5.4**, we discuss why we believe evaluating networks of hazard interactions is important.

## 5.1 Case Study Examples (Nepal and Guatemala)

Networks of hazard interactions are relevant in many locations around the world. Guatemala is an example of a location where multiple different networks of hazard interactions can be identified.  We have identified examples of the wide range of hazards and processes in Guatemala using 21 semi-structured interviews with Guatemalan hazard professionals and personal field observations, during two months of fieldwork in 2014.

- Specific natural hazards: earthquakes, volcanic eruptions, landslides, floods, droughts, tropical storms, extreme
temperatures, subsidence, ground collapse and wildfires.
- Relevant anthropogenic processes: deforestation, inadequate drainage, agricultural practices and building/road construction practices.
- Technological hazards/disasters of relevance: structural collapses, urban fires, chemical pollution and transport accidents.

Specific hazards or processes influencing Guatemala may last for decades (e.g., eruptive activity of Santiaguito, Bluth and Rose, 2004) or days (e.g., Tropical Storm Agatha, Stewart, 2011), impacting large areas (e.g., landslides across 1000s km$^2$, Harp et al., 1981) or small areas (e.g., 20 m ground collapses, Stewart, 2011). A wide range of possible interactions exist in Guatemala between specific natural hazards, anthropogenic processes and technological hazards/disasters. Here we present four case study examples of networks of hazard interactions, with three examples from Guatemala, and one additional example
from Nepal, ordered according to their use in subsequent sections.

i.    *Case Study 1: $M_w = 7.8$ Nepal earthquake, April 2015 [earthquake, landslides, floods].* The 25 April 2015 $M_w = 7.8$ Gorkha earthquake in Nepal impacted several Himalayan nations, triggering a $M_w = 7.3$ aftershock on 12 May 2015 (Bilham, 2015; Collins and Jibson, 2015). The initial earthquake is reported to have triggered 553 aftershocks with $M_w > 4$ in the 45 days after the 25 April 2015 $M_w = 7.8$ Gorkha earthquake (Adhikari et al., 2015). The main shock
and aftershocks rapidly triggered snow avalanches and thousands of landslides, with some of the landslides blocking rivers which in some cases triggered upstream flooding (Collins and Jibson, 2015). The earthquake sequence also increased the probability of further landslides, triggered by subsequent monsoon rains (Bilham, 2015, Collins and Jibson, 2015).

ii.    *Case Study 2: Santiaguito lahars and triggered flooding, Guatemala, approximately annual [volcanic activity, rain,*
*lahars, floods].* In Guatemala, rainfall mobilisation of ash and tephra deposits on active volcanic flanks, such as Santiaguito, frequently result in lahars. These lahars subsequently trigger floods through increased sedimentation, the

addition of large amounts of tephra material to the hydrological system (Harris et al., 2006). This network of hazard interactions (cascades) can be observed on an approximately annual basis, during the rainy season, while Santiaguito is active.

iii. *Case Study 3: Tropical Storm Agatha and eruption of Volcano Pacaya, Guatemala, May 2010 [volcanic activity, tropical storm, landslides, floods, ground collapse, structural collapse].* Tropical Storm Agatha impacted several nations within Central America, hitting the Pacific coastline of Guatemala on 29 May 2010 (Stewart, 2011). It was associated with strong winds and torrential rains (Stewart, 2011; Stewart and Cangialosi, 2012). Within Guatemala, the storm triggered landslides (Wardman et al., 2012) and flooding across much of the Southern Highlands of Guatemala, and contributed to a rare, localised (20 m diameter), rapid-onset ground collapse event (Stewart, 2011) in Guatemala City. The effects of Tropical Storm Agatha in Guatemala were exacerbated by the near-simultaneous eruption of Pacaya, a complex volcano located 30 km southwest of Guatemala City. Pacaya erupted two days prior to the onset of Tropical Storm Agatha on 27 May 2010 (Wardman et al., 2012), ejecting ash and debris across much of Guatemala City. As ash blocked the inadequate drainage system, it increased the intensity of flooding during Tropical Storm Agatha (United Nations, 2010). The combination of fresh ash, volcanic debris and heavy rain, generated lahars (a natural hazard) and structural collapse (a technological hazard/disaster) (Daniell, 2011; Wardman et al., 2012).

iv. *Case Study 4: $M_w = 7.5$ Guatemala earthquake, 1976 [earthquake, ground collapse, landslides, floods].* This $M_w = 7.5$ earthquake triggered multiple aftershocks, and movement on other faults close to Guatemala City (Espinosa, 1976; Plafker et al., 1976). The earthquake triggered some rapid subsidence or ground collapse (Plafker et al., 1976) and more than 10,000 landslides, rock falls and debris flows (Harp et al., 1981). Many of these mass movements occurred along poorly-built road and rail cuttings, blocking vital transport routes (Plafker et al., 1976). Some of the mass movements also blocked rivers and triggered upstream flooding (Plafker et al., 1976; Harp et al., 1981). Breaches of these landslide-dams also resulted in further flooding (Harp et al., 1981).

Many other examples of networks of hazard interactions (cascades) can be observed in the published scientific literature, technical reports, press releases and other forums. It is beyond the scope of this article to compile a comprehensive list of these cascades; however, many can be found in the references noted at the end of this article. We proceed to use the four case study examples outlined above, together with three further theoretical examples, to illustrate two important concepts relating to networks of hazard interactions.

**5.2 Variations in Spatial and Temporal Extent, Frequency and Impact of Networks of Hazard Interactions (Cascades)**

In the example case studies described in **Sect 5.1**, we observe variation in the spatial and temporal extent, frequency and impact of networks of hazard interactions. Networks of hazard interactions (cascades) can vary over many orders of magnitude both spatially and temporally. For example, a tropical storm (lasting several days) may trigger landslides across a small localised

area or an entire region (e.g., Central America). One of these triggered landslides may further block a river causing a small, localised flood or weaken the structural integrity of a dam and cause a large regional flood. We illustrate the wide variation in spatial and temporal extent, frequency of networks of hazard interactions and impacts of such networks using Case Study 1 ($M_w$ = 7.8 *Nepal earthquake and triggered hazards*, April 2015), Case Study 2 (*Santiaguito lahars and triggered flooding, Guatemala, approximately annual*), and Case Study 3 (*Tropical Storm Agatha and eruption of Volcano Pacaya, Guatemala, May 2010*).

In the 2015 $M_w$ = 7.8 'Gorkha earthquake, Nepal' (Case Study 1 in **Sect 5.1**), thousands of landslides were triggered across a wide spatial extent (30,000 km$^2$), with at least 69 of these landslides forming landslide dams (Collins and Jibson, 2015). Many of these dams impounded water, causing flooding, with surface areas ranging from 50 m$^2$ to 35,000 m$^2$ (Collins and Jibson, 2015). Landslides were both triggered in the minutes and days after the earthquake, but also the susceptibility of slopes was changed so as to make landslides more likely in the months to years after the earthquake (Bilham, 2015, Collins and Jibson, 2015)

The regular eruptions of Santiaguito in Guatemala and subsequent lahars/flooding (Case Study 2 in **Sect 5.1**) also illustrate variation across spatial and temporal scales. Volcanic activity may extend over a sub-national, national or multi-national spatial level, and be either short-lived or persist for many decades. The Santiaguito dome in Guatemala, for example, has seen unsteady, extrusive activity since 1922 (Bluth and Rose, 2004), mainly impacting the southwest of Guatemala. Volcanic activity at Santiaguito, in combination with regular rainfall, results in lahars each rainy season which have an impact on the fluvial system at distances of up to 60 km from Santiaguito, including causing flooding (Harris et al., 2006). While in Guatemala in 2014, we confirmed this network of hazard interactions using personal field observations and discussions in seven semi-structured interviews with hazard monitoring and civil protection officials.

Finally, consider the example of Tropical Storm Agatha and the eruption of Volcano Pacaya (May 2010) in Guatemala (Case Study 3 in **Sect 5.1**) which also demonstrates variations in spatial and temporal scale. Tropical Storm Agatha had an impact across multiple nations within Central America (a scale of hundreds of thousands square kilometres). In contrast, one of the secondary hazards associated with this storm was a localised ground collapse event, with a diameter of 20 m (Stewart, 2011).

Networks of hazard interactions (cascades) can also vary in terms of their frequency and impact. For example, they can be observed in low frequency, high-impact events such as the 2015 $M_w$ = 7.8 'Gorkha earthquake, Nepal' (Case Study 1 in **Sect 5.1**). These internationally-publicised events help to raise the profile of networks of hazard interactions (cascades) to an international audience. The 2015 $M_w$ = 7.8 'Gorkha earthquake, Nepal' and resulting secondary hazards resulted in more than 8700 fatalities and 3.5 million people displaced (Bilham, 2015). Networks of hazard interactions (cascades) are also observed in localised, high-frequency events, such as the regular eruptions of Santiaguito in Guatemala and subsequent lahars/flooding (Case Study 2 in **Sect 5.1**). This annual network of interacting hazards (cascades), although not commonly associated with high numbers of fatalities, does have the potential to impact livelihoods of those living in this vicinity and the wider economy

(Harris et al., 2006). During Tropical Storm Agatha (May 2010) in Guatemala (Case Study 3 in **Sect 5.1**), a diversity of impacts included at least nine triggered landslides that caused fatalities (Kirschbaum et al., 2012), as well as the economic costs associated with flooding in Guatemala City and structural collapse caused by the combination of ash and heavy rain (United Nations, 2010; Daniell, 2011; Wardman et al., 2012).

As demonstrated through discussion of these case studies, networks of hazard interactions (cascades) are relevant at diverse spatial and temporal scales, can be both high and low frequency events, and have impacts ranging from fatalities to impacts on livelihoods.

### 5.3 Visualising Networks of Hazard Interactions (Cascades)

Given the prevalence of networks of hazard interactions, we consider here how these networks can be visualised to support multi-hazard assessments of interacting natural hazards. In this section we present two ways of visualising networks of hazard interactions, using Case Study 2 (*Santiaguito lahars and triggered flooding, Guatemala, approximately annual),* Case Study 4 (*1976 $M_w = 7.5$ Guatemala earthquake*) and other hypothetical examples.

In Gill and Malamud (2014), we developed one method of visualising networks of hazard interactions through the use of a 21
× 21 hazard interaction matrix, showing possible interactions between 21 different 'primary' and 'secondary' natural hazards, and then overlaid onto this relevant information about the network of hazard interactions. In **Figure 4** we show this methodology using a hypothetical, but common, example of a network of hazard interactions (cascade) formed exclusively from natural hazards. This network of hazard interactions is presented on a 21 × 21 hazard interaction matrix, which includes both triggered relationships and relationships where one hazard increases the probability of another. It uses a two letter code
for the 21 different natural hazards, the same hazards as those given in **Table 1**. The vertical axis of the matrix in **Figure 4** displays the primary hazards (rows 1 to 21, **EQ** to **IM**). These are the initial hazards that trigger or increase the probability of another hazard occurring. The horizontal axis of the matrix presents these same hazards as potential secondary hazards (columns A to U, **EQ** to **IM**). These are the triggered hazards, or the hazards for which the probability of occurrence has been increased. The 21 hazard types are clustered into six hazard groups, identifiable with different colours as indicated in the key.
Each matrix cell is divided diagonally so that there are two triangles in a cell. Shading in the upper-left triangle of a given cell indicates that the primary hazard could trigger an occurrence of the secondary hazard. Shading in the lower-right triangle of a given cell indicates that the primary hazard could increase the probability of the secondary hazard. It is, of course, possible for both of these triangles to be shaded for any given primary hazard-secondary hazard coupling. The network of hazard interactions (cascade) presented within **Figure 4**, overlaying the 21 × 21 hazard interaction matrix, is a hypothetical example,
initiated by a storm (row 12, **ST**) that triggers flooding (column F, **FL**). This flooding may (row 6, **FL**) subsequently trigger

landslides (column D, **LA**) through the erosion of slope bases. These landslides (row 4, **LA**) could then trigger or increase the probability of further flooding (column F, **FL**) through the damming of rivers.

Using the hazard interaction matrix visualisation framework illustrated in **Figure 4**, we can also represent two of the case study examples introduced in **Sect 5.1**. **Figure 5** shows two examples of networks of hazard interactions (cascades), both from the Southern Highlands of Guatemala. **Figure 5 (top)** visualises some of the hazards and hazard interactions relevant to the 1976 $M_w = 7.5$ Guatemala earthquake (Case Study 4 in **Sect 5.1**). An earthquake (row 1, **EQ**) triggered other earthquakes (column A, **EQ**), landslides (column D, **LA**), and ground collapse (column I, **GC**). The landslides (row 4, **LA**) subsequently blocked rivers and caused flooding (column F, **FL**). **Figure 5 (bottom)** visualises some of the hazards and hazard interactions associated with lahar triggered flooding around the volcano Santiaguito (Case Study 2 in **Sect 5.1**). Heavy rainfall (row 12, **ST**) mobilises volcanic material to trigger lahars (column D, **LA**). These lahars (a form of mass movement) (row 4, **LA**) result in significant volcanic material entering rivers and causing flooding (column F, **FL**).

The hazard/process flow diagram visualisations previously introduced in **Sect. 4** (**Figures 2** and **3**) can also be used to represent complex networks of hazard interactions involving a mixture of natural hazards, anthropogenic processes and technological hazards/disasters. We use the structure of **Figures 2** and **3**, with appropriate replication within the same figure to allow for longer and more complex networks of hazard interactions, and give two theoretical examples (A and B, described further below) in **Figures 6** and **7** of a complex network of hazard interactions. The two hazard/process flow diagram examples in **Figures 6** and **7** show all possible triggering interactions (thick block arrows with solid outlines) and (for simplification) only relevant catalysing/impedance interactions (thin block arrows with dashed outlines). Possible networks of hazard interactions are visualised using light-blue boxes to highlight the relevant hazards/processes (i.e., nodes within a network), and dark-blue arrows to highlight the relevant interactions (i.e., links within a network).

*Theoretical network of hazard interactions, Example A (four links, arrows labelled 1 to 4) using a hazard/process flow diagram.* **Figure 6** shows a primary anthropogenic process catalysing (thin arrow **1**) the triggering relationship between a primary and secondary natural hazard (thick arrow **2**), with the secondary natural hazard then triggering (thick arrow **3**) a primary technological hazard, which in turns triggers (thick arrow **4**) a primary anthropogenic process to occur. An analogous example of this interaction network would be urbanisation increasing overland flow and therefore catalysing (**1**) storm triggered floods (**2**), with the floods then triggering (**3**) an embankment to collapse, which in turn triggers (**4**) anthropogenic drainage and dewatering.

*Theoretical network of hazard interactions, Example B (five links, arrows labelled 1 to 5) using a hazard/process flow diagram.* The network of hazard interactions in **Figure 7** is more complex, with three branches and five interaction relationships highlighted. This example shows a primary natural hazard triggering (thick arrow **1**) a primary technological hazard, which in turn triggers (thick arrow **2**) a primary anthropogenic process. The same primary natural hazard may trigger (thick arrow **3**) a secondary natural hazard. This secondary natural hazard could then trigger (thick arrow **4**) a primary technological hazard and

(thick arrow **5**) tertiary natural hazards. An analogous example of this interaction network would be an earthquake triggering (**1**) a structural collapse, which in turn results in (**2**) increases in infilled (made) ground. The earthquake may also trigger (**3**) landslides, which could trigger (**4**) a road traffic accident and (**5**) flooding.

The overlay of networks of hazard interactions from case studies in **Sect 5.1** on hazard interaction matrices (**Figures 4** and **5**), and the overlay of theoretical examples on hazard/process flow diagrams (**Figures 6** and **7**) can be complemented by other visualisation techniques. For example, when a quantitative evaluation of possible outcomes of interaction relationships is possible, probability trees can be used to assess networks of hazard interactions (e.g., Neri et al., 2008; Marzocchi et al., 2009; Neri et al., 2013). Probability trees are used to visually represent the possible outcomes of an event and add associated probabilities. All three methods are useful for communicating information about specific chains of events. The two visualisation techniques that we have presented here, together with existing probability trees, allow simple and more complex networks of hazard interactions to be evaluated and visualised.

### 5.4 Importance of Networks of Hazard Interactions (Cascades)

We believe that the assessment and visualisation of possible interaction networks (cascades) within multi-hazard methodologies is of importance to both the theoretical and practical understanding of hazards and disaster risk reduction. Here we outline three principal reasons for identifying possible interaction networks.

### 5.4.1 Risk Assessments and Risk Management Benefit by Better Matching Observed Reality

An analysis of past occurrences of hazards and disasters shows that interaction networks are often part of the structure of disasters. The need to better match observed reality, by including interaction networks, is applicable to events of diverse spatial and temporal extent, frequency and impact, as has been discussed in **Sect. 5.2**. The frequency of occurrence of specific networks of hazard interactions demonstrates that more could be done to understand and characterise them. Following the 2015 Gorkha (Nepal) earthquake, the European Geosciences Union (EGU) issued a statement (EGU, 2015) calling for a multi-hazard, integrated approach to risk assessment and the management of natural hazards. This statement also notes the need for agreement within the geoscience community on how to model cascades of natural hazards. This call joins many previous calls (Delmonaco et al., 2007; Kappes et al., 2012; Marzocchi et al., 2012; Gill and Malamud, 2014; Liu et al., 2016) encouraging the assessment of interacting natural hazards, and their integration into multi-hazard methodologies. Assessing interaction networks is therefore important as they are a fundamental part of hazard and disaster events.

### 5.4.2 Changes to Social and Physical Vulnerability During Links of a Multi-Hazard Cascade Event

As one progresses along a network of hazard interactions (cascade), aspects of social and/or physical vulnerability may change following the occurrence of a specific natural hazard, anthropogenic process or technological hazard/disaster. If there is a succession of hazard events (i.e., a network of hazard interactions), there may be progressive changes in vulnerability during

this succession. While some aspects of vulnerability may remain at the same level before and after the occurrence of a specific event, it is also possible that other aspects of vulnerability may increase as pressure is placed on society and infrastructure, thus reducing coping capacity or decrease. Other aspects of vulnerability could also decrease, especially if there are significant time intervals between successive events in a cascade. This could, for example, help facilitate a growth in community awareness and preparation.

This changing vulnerability within a network of hazard interactions can be represented visually, as shown in **Figure 8**, where a series of three hazard events occur in succession and an assumption is made that each hazard event will increase subsequent levels of vulnerability. Before and between these three hazard events, a representation of vulnerability is given, where we illustrate the vulnerability magnitude as proportional to the height of the rectangle. **Figure 8** shows the dynamic nature of vulnerability as one progresses along a network of interacting hazards. In this representation, we have assumed that there are increases in vulnerability as the chain of events progresses, but we note that this will not always be the case. On the ground these changes to social and physical vulnerability may be observed in different ways. For example, buildings may have sustained significant damage so that they are more likely to collapse during an aftershock. Hospitals may be at maximum capacity following an earthquake and therefore not able to respond effectively if a subsequent typhoon results in further casualties. Injuries sustained by a community during an earthquake may mean that they have a reduced capacity to evacuate if a tsunami is subsequently triggered.

These examples demonstrate that existing assessments of vulnerability may rapidly become out of date following a hazard event. The identification of possible interacting hazard networks in a given region would allow improved planning of possible changes in vulnerability during successive events. In turn, this could help to improve preparedness efforts.

**5.4.3 Allocation of Resources for Disaster Risk Reduction**

In addition to the risk reduction benefits that come from the last two points, understanding how chains of interacting hazards are initiated and propagated may help determine how to invest resources to minimise disruption should a specific network of interacting hazards occur. Scientific and management efforts can be focused on (i) preventing the initiation of interaction networks and (ii) reduce or eliminate specific interactions along the interacting hazard network. It may not always be possible to prevent an initial primary hazard from occurring, but sensible investments in structural and non-structural mitigation measures may reduce the likelihood of specific networks of hazard interactions propagating. While we cannot currently prevent a tropical storm from forming and hitting land, for example, measures may be taken to improve drainage and reduce flooding, reinforce certain slopes that are susceptible to failure, or improve urban management to reduce structural collapses, urban fires and water contamination.

**6 Conclusions**

In this research and commentary article, we have sought to advance the understanding of enhanced multi-hazard frameworks, which we believe to be of relevance to improved Earth-systems management. We advocate an approach that goes beyond multi-layer single hazard approaches to also encompass interaction relationships and networks of interactions (cascades). This study has described this integrated approach, noting that to do otherwise could distort management priorities, increase vulnerability to other spatially relevant hazards or underestimate risk. The development of an enhanced framework to assess and characterise interactions and networks of interactions first required a description of three principal groups of hazards/processes, including natural hazards, anthropogenic processes and technological hazards/disasters. These three groups can interact in a range of different ways, with three interaction relationships discussed in the context of this article: triggering relationships, increased probability relationships, and catalysis/impedance of other hazard interactions. In addition to those circumstances where one stimulus triggers one response, it is highly likely that more than one of these interactions can be joined together to form a network of interactions, chain or cascade event. We have developed enhanced frameworks to visualise in two different ways (hazard/process flow diagrams in **Figures 2**, **3**, **6** and **7,** and hazard interaction matrices in **Figures 4** and **5**) these interactions and networks of interactions (cascades). These frameworks, visualisations and associated commentary:

    i.    Reinforce the importance of enhanced multi-hazard approaches, integrating hazard interactions and networks of interactions to better model observed dynamics of the Earth system.

    ii.    Offer a more holistic approach to assessing hazard potential, helping to improve management of those aspects of the Earth system that are relevant to disaster risk reduction.

    iii.    Support the research community to consider future research directions in the context of multi-hazard research in regional settings.

Better characterisation and integration of interactions and networks of interactions into multi-hazard methodologies can contribute to an improved theoretical and practical understanding of hazards and disaster risk reduction.

**Acknowledgements**

This research was funded by a studentship grant from NERC/ESRC grant: NE/J500306/1. The authors wish to thank INISVUMEH (Guatemala) and CONRED (Guatemala) for their assistance in the field, when reviewing case studies. The authors also wish to thank Reik Donner (Potsdam Institute for Climate Impact Research, Germany) and one anonymous reviewer for their helpful and constructive reviews, and Christian Franzke for his support as editor for this paper.

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

**Table 1.** Examples of hazard/process types, grouped into three categories: Natural Hazards (classification of 21 hazards from Gill and Malamud, 2014), Anthropogenic Processes and Technological Hazards/Disasters.

| Hazard/Process Group | Examples |
| --- | --- |
| Natural Hazards | Earthquake, tsunami, volcanic eruption, landslide, snow avalanche, flood, drought, regional subsidence, ground collapse, soil (local) subsidence, ground heave, storm, tornado, hailstorm, snowstorm, lightning, extreme temperature (hot and cold), wildfire, geomagnetic storm, impact event. |
| Anthropogenic Processes | Groundwater abstraction, subsurface mining, subsurface construction, fluid injection, vegetation removal, urbanisation, surface mining, drainage and dewatering, reservoir construction, wastewater injection, chemical explosion. |
| Technological Hazards/Disasters | Structural collapse, nuclear reactor failure, urban fire, chemical pollution, dam collapse, industrial explosion, transport accident. |

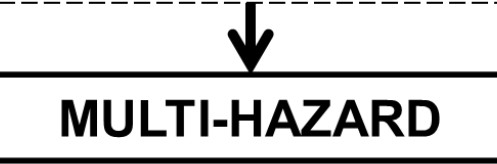

**MULTI-LAYER SINGLE HAZARD**

(1) *Hazard Identification and Comparison.* The identification and valid comparison of all identified individual hazards, relevant to a given spatial region.

(2) *Hazard Interactions.* The identification and characterisation of all possible interactions between identified hazards.

(3) *Hazard Coincidence.* An investigation into the impacts of two or more hazards coinciding spatially and/or temporally, such that the hazard potential and/or vulnerability may differ from the sum of its parts.

(4) *Dynamic Vulnerability.* An understanding of how one, or a series of hazards, will impact upon the vulnerability and resilience of a community, thus changing the overall future risk to a location or community.

**MULTI-HAZARD**

**Figure 1**. Multi-hazard risk framework (from Gill and Malamud, 2014). Shown is the progression from a multi-layer single hazard approach to a full multi-hazard risk approach that includes: (i) hazard identification and comparison, (ii) hazard interactions, (iii) spatial/temporal coincidence of natural hazards, and (iv) dynamic vulnerability to multiple stresses (when progressing from the assessment of hazard to the assessment of risk).

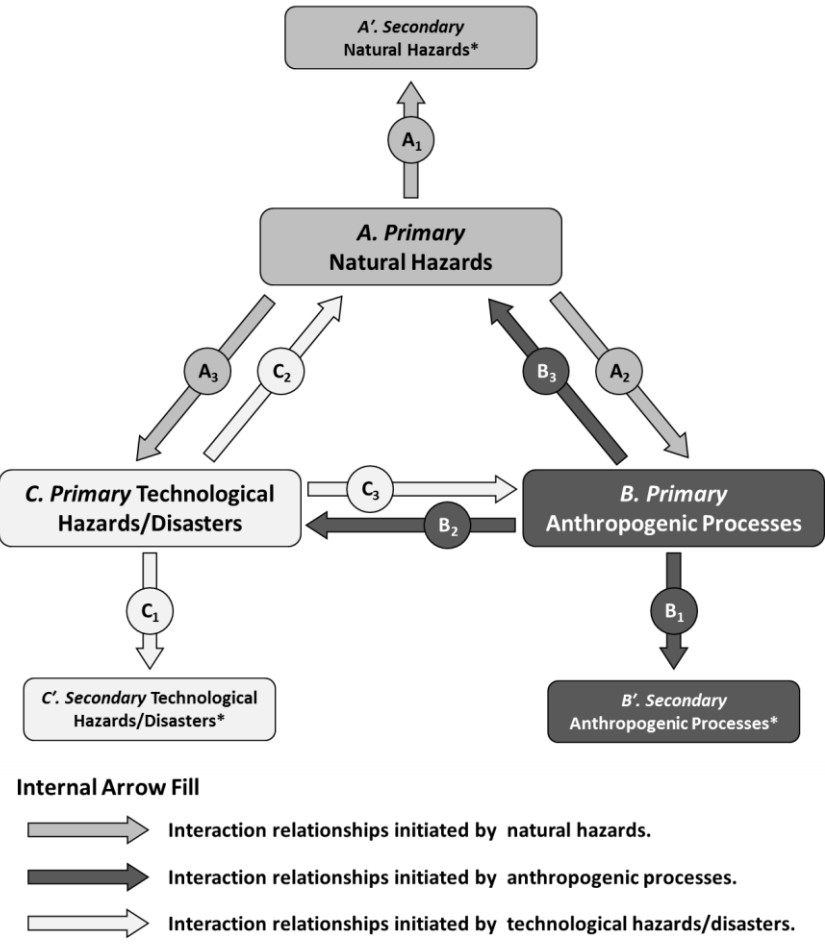

**Internal Arrow Fill**

➡ Interaction relationships initiated by natural hazards.

➡ Interaction relationships initiated by anthropogenic processes.

➡ Interaction relationships initiated by technological hazards/disasters.

*Secondary hazards may initiate further interactions.*

**Examples for Interaction Relationships A₁–C₃**

|   | 1 | 2 | 3 |
|---|---|---|---|
| **A** | Earthquake triggers landslides | Drought triggers agricultural practice change | Earthquake triggers nuclear reactor meltdown |
| **B** | Subsurface mining drives dewatering | Subsurface mining triggers mine collapse | Fluid injection triggers earthquakes |
| **C** | Transport accident triggers chemical spill | Dam collapse triggers flooding | Structural collapse drives infilled (made) ground |

**Figure 2**. Interaction relationships (triggering) framework using a hazard/process flow diagram. A framework for hazard/process interactions is given here, which highlights triggering relationships between three groups: (**A**) natural hazards, (**B**) anthropogenic processes and (**C**) technological hazards/disasters. Arrows are used to illustrate interaction relationships,

with the arrow fill colour indicating the 'source' or initiation of the trigger (medium grey: natural hazards; dark grey: anthropogenic processes; light grey: technological hazards/disasters). We use a prime ($A'$, $B'$, $C'$) to indicate secondary hazards/processes triggered by the same primary hazard/process group ($A$, $B$, $C$). Arrows are labelled ($A_1$–$A_3$, $B_1$–$B_3$, $C_1$–$C_3$) according to the hazard or process type of the 'trigger source' (i.e., $A$, $B$, $C$), and followed by sequential subscript numbering.

5 Numbering starts ($A_1$, $B_1$, $C_1$) with the triggering relationship between the same primary and secondary hazard or process type (e.g., a primary natural hazard triggering a secondary natural hazard) and progresses clockwise. Examples of each interaction are given in the table at the bottom of the figure, where the vertical axis indicates the source of the primary hazard/process ($A$, $B$, $C$), and the horizontal axis indicates which subscript is being referred to ($1$–$3$).

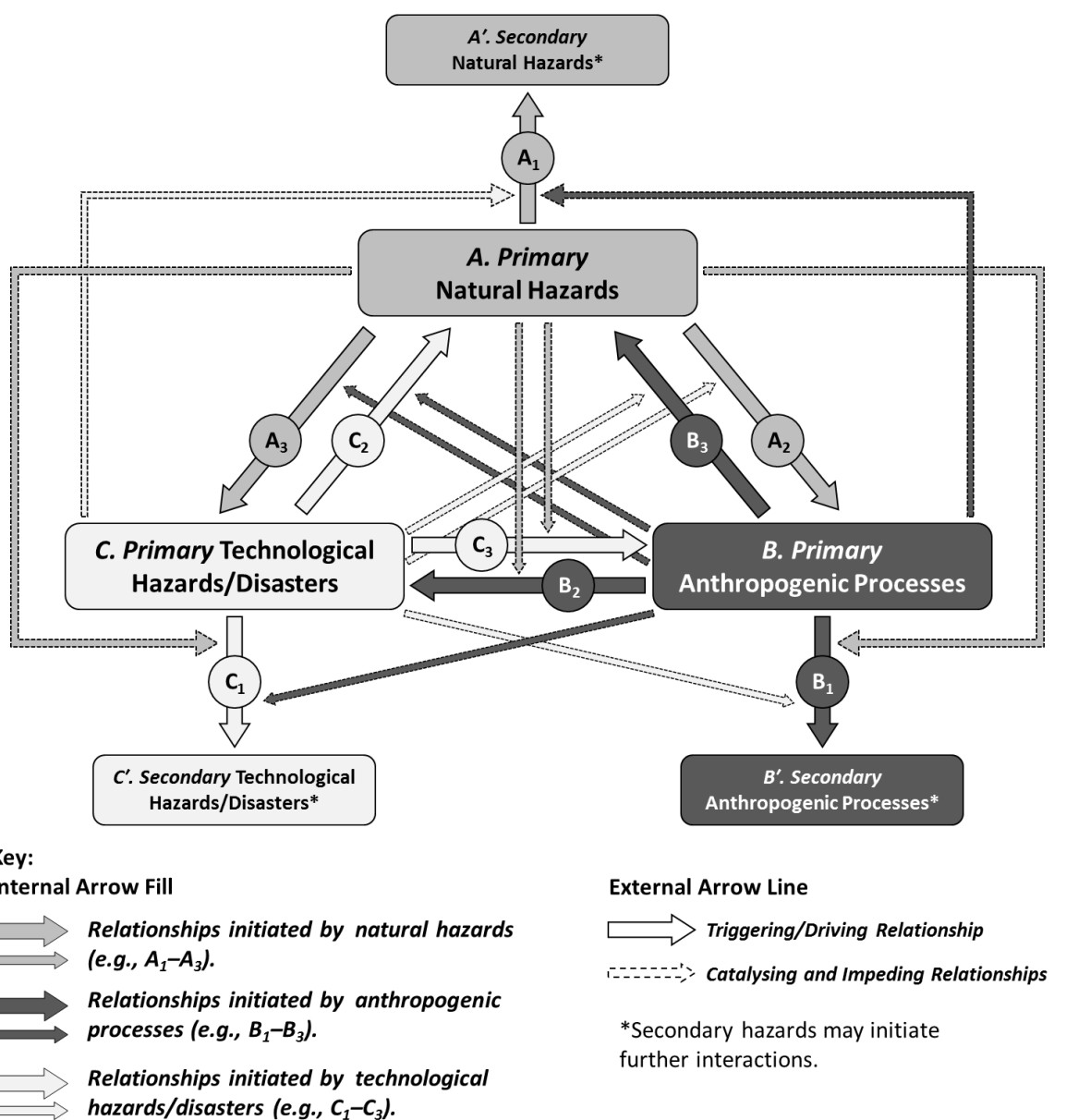

**Figure 3**. Interaction relationships (triggering, catalysing and impeding) framework using a hazard/process flow diagram. Interactions in the form of triggering relationships (**Figure 2**), and catalysing/impedance interactions are possible between (**A**) natural hazards, (**B**) anthropogenic processes and (**C**) technological hazards/disasters. We use a prime (**A′**, **B′**, **C′**) to indicate secondary hazards/processes triggered by the same primary hazard/process group (**A**, **B**, **C**). We contrast here triggering relationships (thick block arrows with solid outlines) and catalysing/impedance relationships (thin block arrows with dashed outlines). The internal arrow fill colour indicates the group of hazards or processes to which the catalyst/impeder belongs

(medium grey: natural hazard; dark grey: anthropogenic process; light grey technological hazard/disaster). Descriptions of arrow labels ($A_1$–$A_3$, $B_1$–$B_3$, $C_1$–$C_3$) can be found in **Figure 2** caption. Examples of catalysing and impedance relationships are given in **Sect. 4.4**.

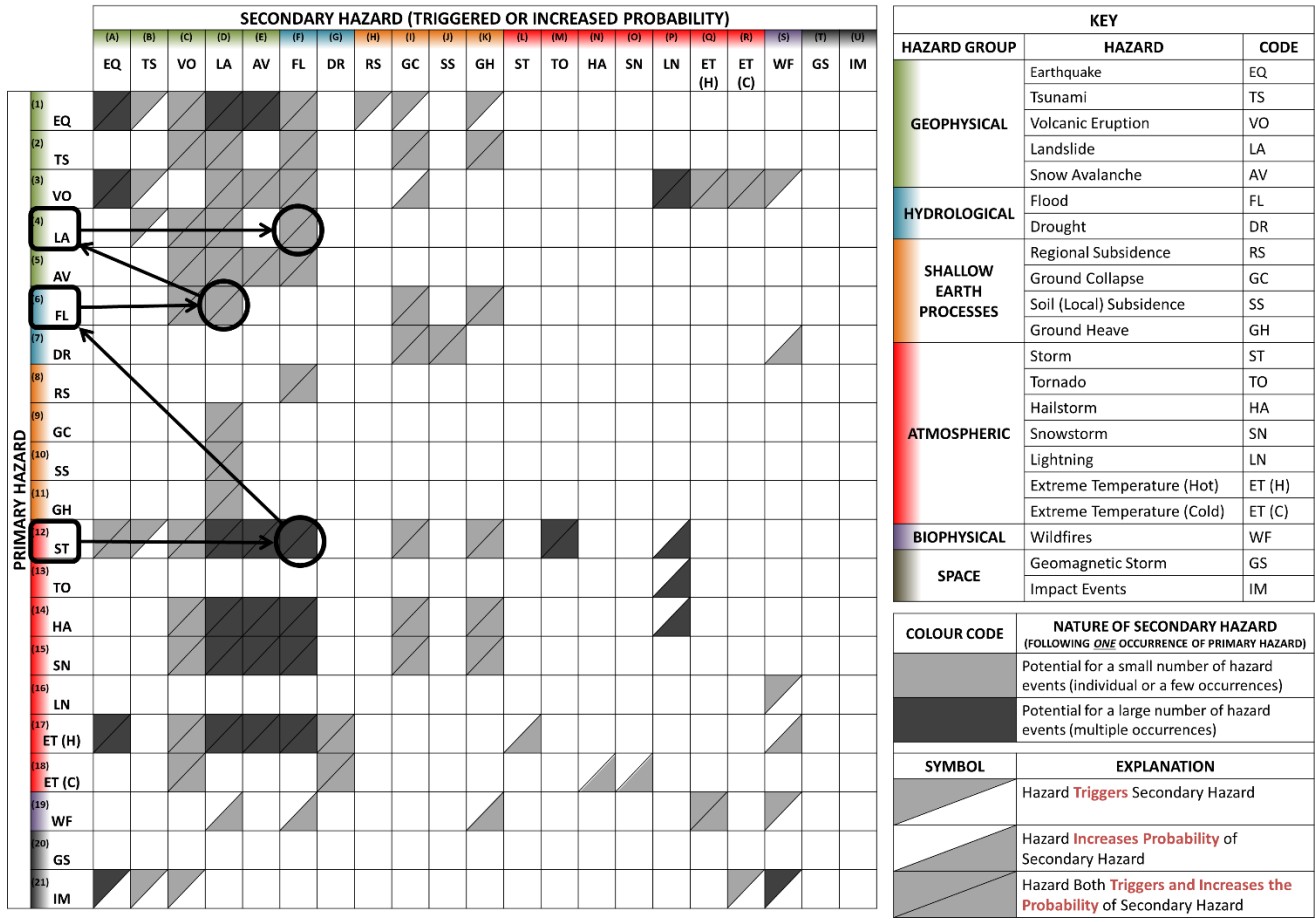

**Footnotes**

**[1A,D,E; 3A,P; 12D-F,M,P; 13P; 14D-F,P; 15D-F; 17A,D-F; 21A]** The secondary hazards in these cases are all accepted to most likely occur as large numbers of events, and are thus analysed in this way.

**[1C]** There is disagreement in the literature about the nature of this relationship .

**[2,6,12,14,15C]** Water input triggers or increases the probability of a phreatic/phreatomagmatic eruption.

**[3I]** Volcanism increases the acidity of rain, promoting dissolution of carbonate material.

**[12A]** Low pressure systems have been shown to trigger or increase the probability of slow earthquakes on faults that are already close to failure (Liu *et al.,* 2009).

**[17A,C-F]** Secondary hazards triggered or have an increased probability over a range of time-scales, through snow and glacial melting.

**[18C]** Long term reductions in temperature can increase glaciation and thus decrease sea-levels. This reduction in sea-levels can reduce confining pressures, promoting volcanic eruptions.

**Figure 4**. An example of a network of hazard interactions (a cascade system) (from Gill and Malamud, 2014) using a hazard interaction matrix. A 21 × 21 matrix with primary natural hazards on the vertical axis and secondary hazards on the horizontal axis. These hazards are coded, as explained in the key. This matrix shows cases where a primary hazard could trigger a secondary hazard (upper-left triangle shaded) and cases where a primary hazard could increase the probability of a secondary hazard being triggered (bottom-right triangle shaded). Where both triangles are shaded, this indicates that the primary hazard could both trigger and increase the probability of a secondary hazard. Also distinguished are those relationships where a primary hazard has the potential to trigger or increase the probability of multiple occurrences of the secondary hazard (dark

grey), and few or single occurrences of the secondary hazard (light grey). Hazards are grouped into geophysical (green), hydrological (blue), shallow Earth processes (orange), atmospheric (red), biophysical (purple) and space/celestial (grey). Footnotes give further information about some of the relationships. This matrix can be used to present an example of a network of hazard interactions (cascade). In this network of hazard interactions example (illustrated using rectangles, circles and arrows overlaid on the matrix), a storm event (**ST**) triggers flooding (**FL**), which then triggers landslides (**LA**). These landslides (**LA**) may then trigger or increase the probability of further flooding (**FL**) through the blocking of a river or the increase of sediment within the fluvial system.

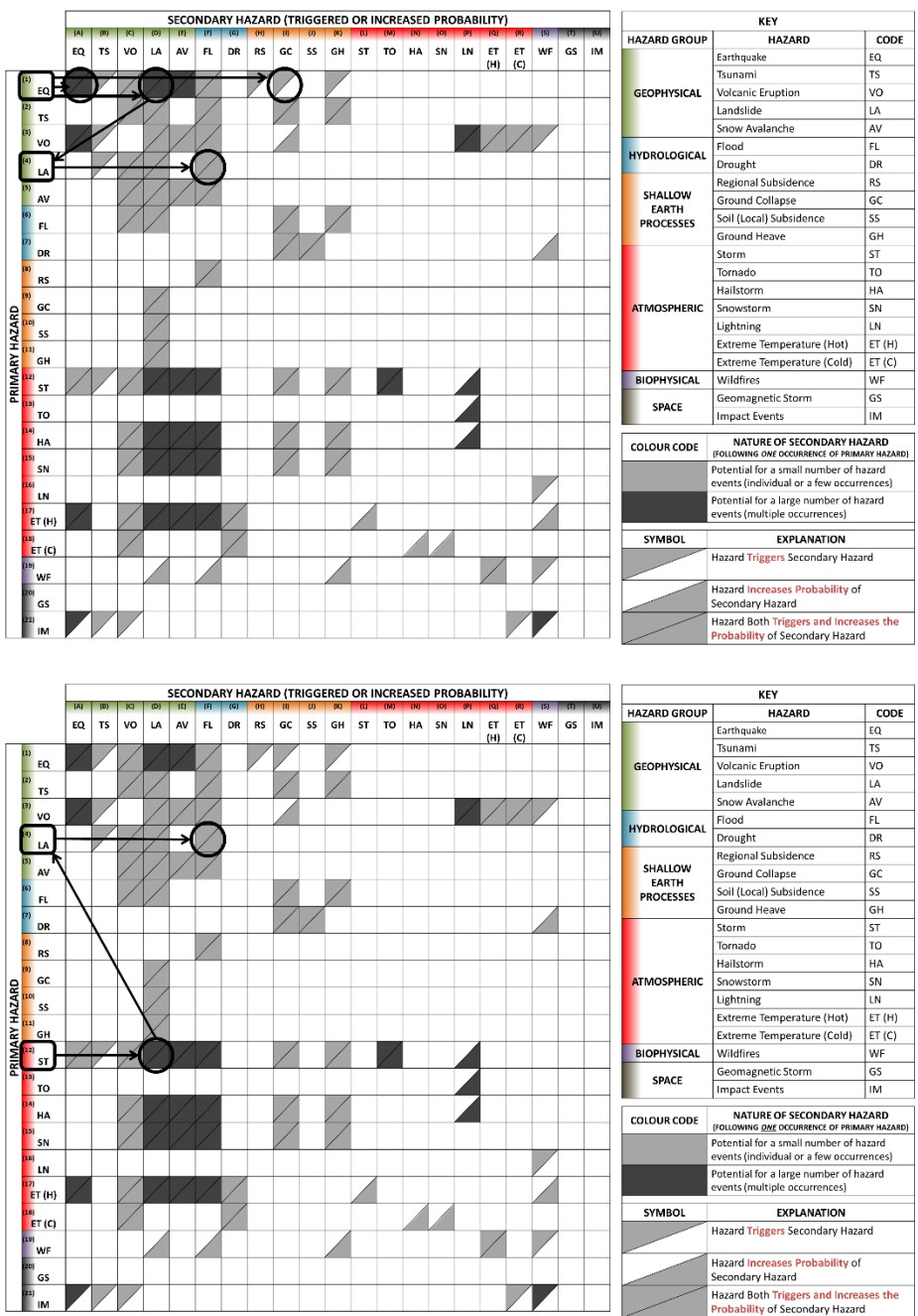

**Figure 5**. Two examples of networks of hazard interactions (cascade systems) using a hazard interaction matrix. Hazard interaction networks based on (*top*) the 1976 Guatemala earthquake sequence, and (*bottom*) lahar triggered flooding associated with Santiaguito, Guatemala. Both network examples are place on a 21 × 21 hazard interaction matrix, adapted from Gill and Malamud (2014), and described in detail within the caption of **Figure 4**. In the top example (described in **Sect. 5.2**), based on

information from Espinosa (1976), Plafker et al. (1976) and Harp et al. (1981), we use rectangles, circles and arrows to illustrate the network of hazard interactions for an earthquake (**EQ**) triggering further earthquakes (**EQ**), landslides (**LA**) and rapid subsidence/ground collapse (**GC**). The Landslides (**LA**) were then noted to have blocked rivers, causing flooding (**FL**). The bottom network of hazard interactions example (also described in **Sect. 5.2**), is based on information from Harris et al. (2006) and confirmed by personal field observations and seven semi-structured interviews with hazard monitoring and civil protection officials while the authors were in Guatemala in 2014. The bottom example shows (again using rectangles, circles and arrows) rain storms (**ST**) triggering lahars (**LA**) on the flanks of Santiaguito. These lahars enter the hydrological system and result in flooding (**FL**) downstream.

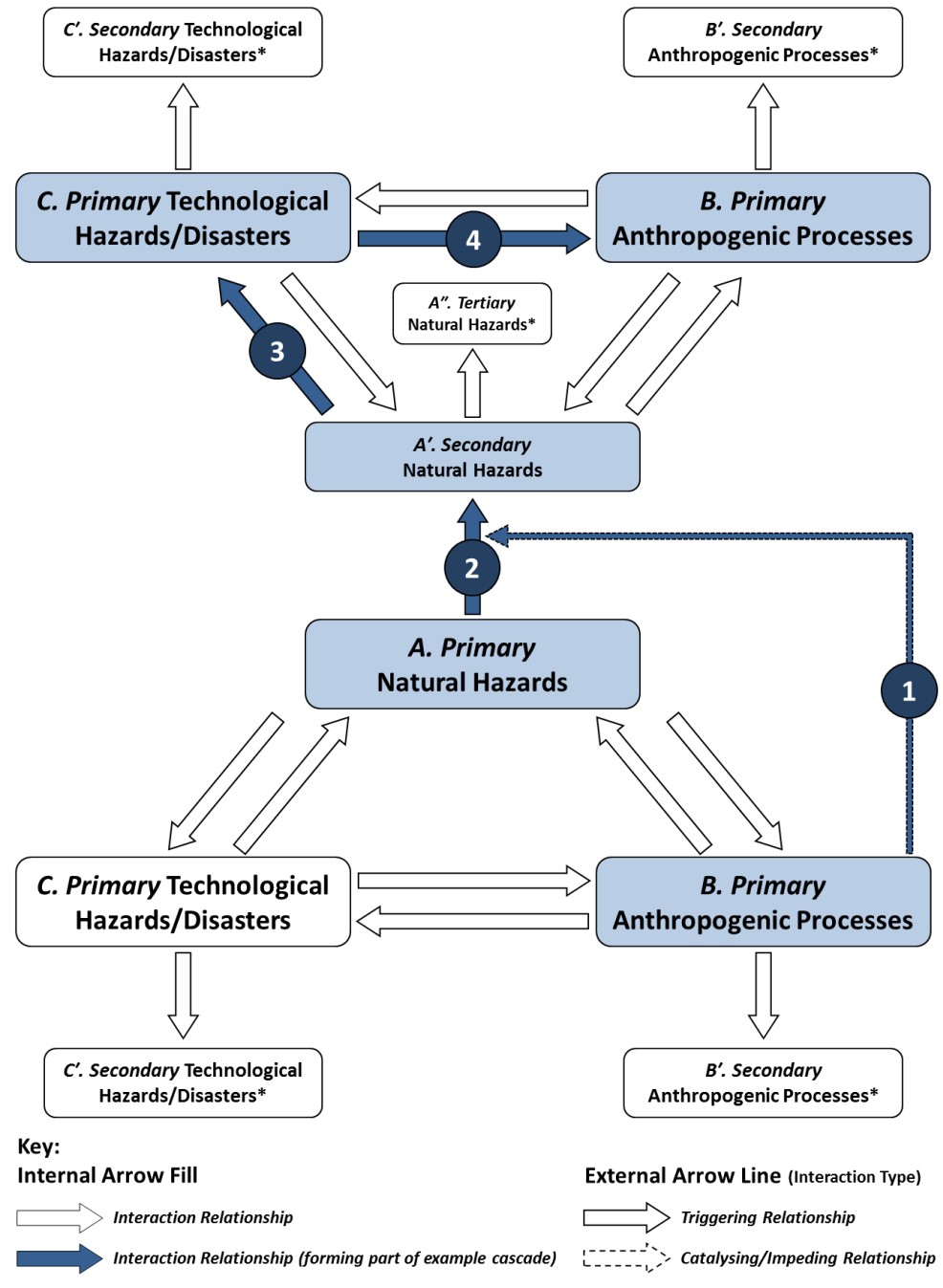

**Figure 6**. Network of hazard interactions (Example A) using a hazard/process flow diagram. Using the visualisation frameworks constructed in **Figures 2** and **3**, an example of an interaction network (cascade) can be presented. Three hazard/process groups are included: (**A**) natural hazards, (**B**) anthropogenic processes and (**C**) technological hazards/disasters.

Arrows are used to illustrate interaction relationships, with both triggering relationships (thick block arrows with solid outlines) and relevant catalysing/impedance relationships (thin block arrows with dashed outlines). For clarity of communication, those catalysing/impedance relationships not of relevance to the specific example are not included. See **Figures 2** and **3** caption explanations for further details. Arrows within the example network of hazard interactions are labelled (**1**–**4**) and shaded dark blue to highlight the relevant pathway. In this example, a primary anthropogenic process catalyses (**1**) the triggering relationship between a primary and secondary natural hazard (**2**), with the secondary natural hazard then triggering (**3**) a primary technological hazard, which in turns triggers (**4**) a primary anthropogenic process to occur.

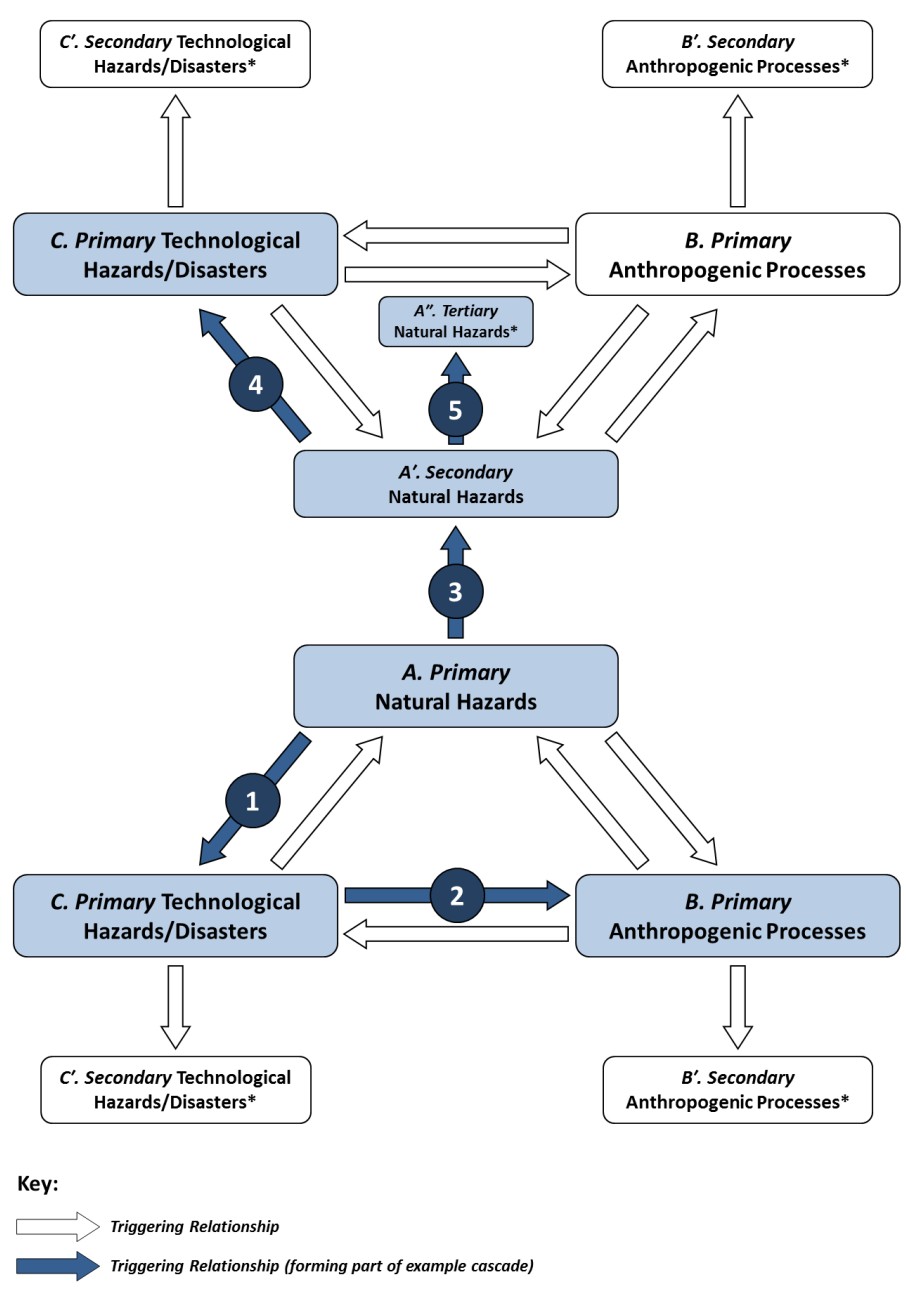

**Figure 7**. Network of hazard interactions (Example B) using a hazard/process flow diagram. Using the visualisations constructed in **Figures 2** and **3**, an example of an interaction network (cascade) can be presented. In this example the network is more complex than in Example A (**Figure 6**), with three branches and five interaction relationships highlighted here. Three

5    hazard/process groups are included: (**A**) natural hazards, (**B**) anthropogenic processes and (**C**) technological hazards/disasters.

Arrows are used to illustrate interaction relationships, with both triggering relationships (thick block arrows with solid outlines) and relevant catalysing/impedance relationships (thin block arrows with dashed outlines). For clarity of communication, those catalysing/impedance relationships not of relevance to the specific example are not included. See **Figures 2** and **3** caption explanations for further details. Arrows within the example network of hazard interactions are labelled (**1**–**5**) and shaded dark blue to highlight the relevant pathway. This example shows a primary natural hazard triggering (**1**) a primary technological hazard, which in turn triggers (**2**) a primary anthropogenic process. The same primary natural hazard may trigger (**3**) a secondary natural hazard. This secondary natural hazard could then trigger (**4**) a primary technological hazard and (**5**) tertiary natural hazards.

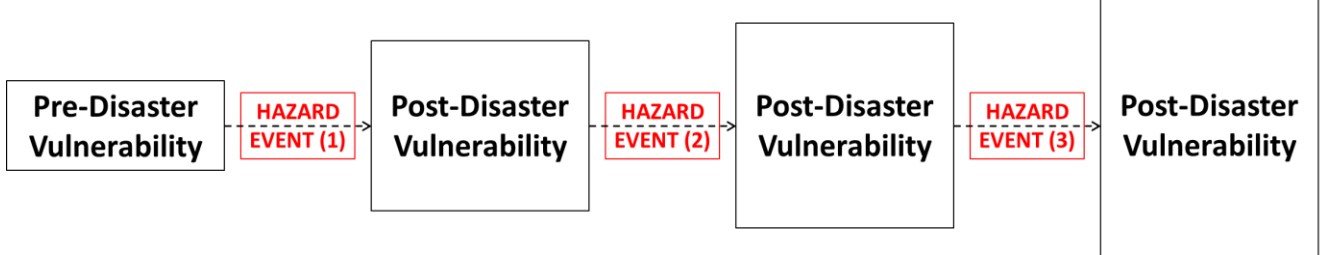

**Figure 8**. Example of vulnerability changes within a network of hazard interactions (cascade). A representation of changing vulnerability during a hazard cascade, where the magnitude of vulnerability is proportional to the size of the box. Following a disaster event, pressures on society, infrastructure and coping capacity are likely to be increased, and thus the vulnerability of

5    a community and its systems/assets to further shocks or hazards may increase.