# Peer review of "Hazard Interactions and Interaction Networks (Cascades) within Multi-Hazard Methodologies"

_Earth System Dynamics, 2015_

## Referee Comment (RC1) · R. V. Donner (Referee) · 25 Feb 2016

Gill and Malamud discuss a general framework for extending the commonly taken view on hazard assessment by taking interdependences of different types linking various hazards into account. Since such interdependences may significantly affect the susceptibility of regions to hazards of different types, I fully agree with their basic argument that hazard interactions need to be carefully considered for obtaining qualitative and quantitative risk assessments. The discussion presented by the authors is scientifically sound and does not only highlight the insufficiencies of previous approaches, but also proposes possible ways to reaching such realistic assessment. The latter aspect appears to be the main achievement of the present work that goes beyond what has

been discussed by the authors in their recent review paper (Gill and Malamud, 2014). At the present stage, this work is purely qualitative, and future studies describing its further extension towards quantitative multi-hazard risk assessments would be most welcome. To this end, this excellent work can be considered as an important milestone on this way.

While the manuscript already presents excellent work, I would like to recommend additionally paying specific attention to some minor points listed below to possibly make it even better:

1. Regarding the anthropogenic processes briefly discussed in Section 3.2, I think that another possibly important example of such processes would be induced seismicity, i.e., seismic activity resulting from human activities like construction of large artificial reservoirs or mining/drilling activities. Recent corresponding references on possible cases of indiced seismicity include, among others, Kerr and Stone, Science 323, 322, 2009, or Hough and Page, Bulletin of the Seismological Society of America, doi: 10.1785/0120150109. I would like to suggest adding some brief comment on such anthropogenic processes.

2. In Section 3.3, the authors state that "similarities exist between the routine hazard events of technology and the definition of anthropogenic processes". Frankly speaking, I do not clearly understand the exact differentiation between both types of hazards from the present discussion. Providing some explicit examples highlighting this difference could be helpful to clarify this point.

3. Regarding their discussion of interaction relationships, I have got the impression that the authors consider such relationships as exclusively unidirectional. Or put differently: the present framework is developed and illustrated with the (in my opinion, unnecessary) implicit restriction of unambiguous cause-effect relationships among hazards (as manifested in the terminology of primary versus secondary hazards). In this regard, I was wondering if there exist any examples of bidirectional interactions, either associated with their simultaneous occurrence or relating to feedbacks. I would be keen to learn about the authors' opinion on this question.

4. In Section 4.1, the authors describe triggering relationships as "causal" relationships (as opposed to the other two types of interactions among hazards). However, depending on which specific notion of causality is considered, I would argue that the other two types also denote some "causal" (in the sense of directional) relations, yet of different quality than triggering relationships where this relation is "most direct".

5. The authors emphasize (p.7, l.28) that triggering interactions can occur between a diverse range of hazards and processes. I think that also the issue of scales (both temporal and spatial) is something important to consider when aiming to characterize triggering relationships. It might be interesting to elaborate further on this aspect.

6. Throughout Section 4, there is considerable redundancy between figure captions and the main text (e.g., between the last paragraph of Section 4.1 and the caption of Figure 2). I think that this redundancy should be reduced.

7. In Section 4.2, the authors state that increased probability relationships "change the frequency or extent of the secondary hazard or process". When considering hazards in some abstract (probabilistic / return period) sense, I agree with this. However, at the level of individual events, a specific primary event would rather change the proximity (and specific characteristics) of the specific next secondary hazard to come affected by the increased probability relationship. From the present paper, it is not always completely clear if the authors wish to consistently take the probabilistic or the individual event viewpoint (not necessarily contradicting each other in general).

8. In Section 4.3, the authors contrast interaction relationships that "include large numbers of frequently-occurring interactions" and such "that are considered to be less common". This seems to me a rather arbitrary distinction (i.e., there are no two distinct types, but rather a continuum of cases according to the frequency of interactions). I was wondering if the authors have any objective criteria for associating catalysing and

impedence relationships to any of the two aforementioned types.

9. In Section 5, the authors first refer to four case studies, later to two. The first two cases detailed in Section 5.1 are not used any further in the following text, while the other two cases are exclusively discussed in the visualization context. All four examples are clearly relevant in the context of this work, but the way they are introduced and detailed in different parts of the manuscript appears slightly confusing.

Despite not being a native speaker, I was wondering about a few words and kindly ask the authors to cross-check them:

* p.3, l.26: "selected"

* p.9, l.21: "...processes relevant interactions can be..."

* p.10, l.15: "evaluating networks... is important"

* p.15, l.27: The meaning of the term "Earth-systems management" is not fully obvious to me.

---

## Referee Comment (RC2) · W. Marzocchi (Referee) · 27 Feb 2016

The paper discusses the importance of modeling the interactions among different hazards, natural and not, to obtain a more realistic risk assessment. The paper is well written (maybe with some unnecessary repetitions), and the topic of great interest. My general opinion is positive but I think that the paper misses to take into account some key issues for a proper multi-hazard assessment.

The first issue, and probably the most important, is that the authors do not discuss the forecasting time windows for hazard assessment. The importance of this topic for multi-hazard and multi-risk assessment has been already discussed in Marzocchi et al (2012; W. Marzocchi, A. Garcia-Aristizabal, P. Gasparini, M.L. Mastellone, A. Di

Ruocco, 2012. Basic principles of multi-risk assessment: a case study in Italy. Natural Hazards, 62, 551-573), and it will be summarized here. The 'hazard' is essentially the probability of a threatening event in one specific time-space window. The (forecasting) time window is of critical importance and it is usually related to the specific use of the hazard assessment in terms of risk reduction. For instance, a long-term hazard assessment (e.g., a forecasting time window of 50 years) is usually adopted for land use planning, like to define the building code for the earthquakes threat. On the other hand, the management of rapidly evolving emergencies usually requires short-term assessments (e.g., days to weeks). Considering long- or short-term hazard provides completely different scenarios for hazard interactions. For example, let us consider the landslide hazard. We can calculate the probabilities of landslides occurrence for the next decades just simply looking at the historical catalog. The fact that most of landslides are caused by earthquakes is not relevant in the long-term hazard because this is true also for the landslides occurred in the past and reported in the historical catalog (assuming that the long-term earthquake rate is not changing with time). So, the interaction between earthquakes and landslides could be irrelevant for the long-term hazard assessment. Of course, if we are considering the short-term hazard assessment (as I guess the authors are doing), the fact that a large earthquake (like the Gorkha event) has just occurred modifies significantly the probability of landslides in the next few months (due to the occurrence of aftershocks). So, the interaction between hazards has to be considered specifying clearly the forecasting time window. Marzocchi et al (2012) explore in detail this point showing a probabilistic framework to explain when hazard interactions are relevant or not. (see section 3.2).

The second issue is that the authors seem to confuse the hazard with the risk. Hazard is an essential component of risk assessment, but it is not the risk. So, I think that introducing vulnerability interactions in the multi-hazard assessment is not proper (like they did in their figure 1). Again, Marzocchi et al (2012) addressed this distinction in their Figure 2. I emphasize that this distinction is not only semantic. In a multi-hazard/risk perspective, it is important to mention that hazards cannot be (usually) compared,

while the risks can. In fact, hazards are usually represented by curves that represent the exceedance probability of some quantity of interest, like ash fall thickness, ground shaking acceleration, etc. So it is not clear to me how we can compare them meaningfully and say when an hazard is higher than another one. On the other hand, the associated risks can be easily compared.

The final issue is about the distinction between "triggered" and "increase probability" made by the authors. This distinction is not clear to me. It seems that the authors assume that triggering is a deterministic link between hazards. In my view the concept of triggering and of increase probability are identical, at least this is true in seismic and volcanic hazard. We can talk of deterministic causal relationship only retrospectively (when you have already observed that one event triggered another one). But when we see the problem in a prospective way, the occurrence of an event *may* increase the probability to trigger another event, but very rarely we can be sure about this triggering. Of course the authors can mean something different, but I think that this distinction has to be clarified.

As a final suggestion, I would like to see a little bit more emphasis on the probabilistic and quantitative nature of hazard assessment. This is just a suggestion and it is not mandatory, but I do think that quantitative assessment of the hazard (and of the consequent risk) is the only possible strategy to plan rational risk reduction actions.

To sum, I am positive about this paper because it reiterates the importance of the interaction among different hazards in a multi-risk perspective. This is certainly commendable. However, I think that the authors should make an effort to address the points raised above.
* * *

---

## Author Comment (AC1) · 14 Mar 2016

**Initial Response to Reviewer Comments**

**Title: Hazard Interactions and Interaction Networks (Cascades) within Multi-Hazard Methodologies**

**[doi:10.5194/esd-2015-94] submitted to Earth System Dynamics**

We thank Reviewer #1 (Reik Donner) and Reviewer #2 (Warner Marzocchi) for their positive, 5 insightful and constructive comments on our manuscript, and for waiving their right to anonymity.

In order to support the interactive discussion nature of the review process, we have drafted some initial responses [highlighted grey] to each of their comments, presented in the subsequent pages. We believe the comments made by both reviewers are fair and will improve our overall argument, and thank both Reik Donner and Warner Marzocchi for their time in helping us to create a paper with a

10 clearer argument.

We look forward to any further comments from the reviewers or editor. We would be delighted to then prepare and submit a revised manuscript based on these comments and any further advice from the editor.

Joel C. Gill and Bruce D. Malamud

15 (Department of Geography, King's College London, UK, corresponding author: joel.gill@kcl.ac.uk)

**REVIEWER #1 (COMMENTS TO AUTHOR): Reik Donner**

- **20 General comments:** Gill and Malamud discuss a general framework for extending the commonly taken view on hazard assessment by taking interdependences of different types linking various hazards into account. Since such interdependences may significantly affect the susceptibility of regions to hazards of different types, I fully agree with their basic argument that hazard interactions need to be carefully considered for obtaining qualitative and quantitative risk assessments. The discussion presented by the authors is scientifically sound
- 25 and does not only highlight the insufficiencies of previous approaches, but also proposes possible ways to reaching such realistic assessment. The latter aspect appears to be the main achievement of the present work that goes beyond what has been discussed by the authors in their recent review paper (Gill and Malamud, 2014). At the present stage, this work is purely qualitative, and future studies describing its further extension towards quantitative multi-hazard risk assessments would be most welcome. To this end, this excellent work
- 30 can be considered as an important milestone on this way. While the manuscript already presents excellent work, I would like to recommend additionally paying specific attention to some minor points listed below to possibly make it even better.

Authors: We are very grateful for this positive, constructive and helpful summary, recognising the value of 35 this contribution to the literature on multi-hazard assessments. We have attempted to address each of the points that you have listed below, and appreciate your thoughtful interaction with our manuscript. We have added some clarifying words at the beginning of each of your comments in [] to give a highlight of what the point is about.

- 40 1.[Induced seismicity as an example of anthropogenic processes] Regarding the anthropogenic processes briefly discussed in Section 3.2, I think that another possibly important example of such processes would be induced seismicity, i.e., seismic activity resulting from human activities like construction of large artificial reservoirs or mining/drilling activities. Recent corresponding references on possible cases of induced seismicity include, among others, Kerr and Stone, Science 323, 322, 2009, or Hough and Page, Bulletin of the
- 45 Seismological Society of America, doi: 10.1785/0120150109. I would like to suggest adding some brief comment on such anthropogenic processes.

**Authors**: We agree that induced seismicity is an important example of anthropogenic processes resulting in the triggering of a natural hazard, in this case seismic activity. Following the helpful suggestion, we will include induced seismicity in a revised version of **Section 3.2**, introducing the following additional papers to the manuscript:

- Deng, K., Zhou, S., Wang, R., Robinson, R., Zhao, C., & Cheng, W. (2010). Evidence that the 2008 Mw 7.9 Wenchuan earthquake could not have been induced by the Zipingpu Reservoir. *Bulletin of the Seismological Society of America*, **100**(5B), 2805-2814.
  - González, P. J., Tiampo, K. F., Palano, M., Cannavó, F., & Fernández, J. (2012). The 2011 Lorca earthquake slip distribution controlled by groundwater crustal unloading. *Nature Geoscience*, **5**(11), 821-825.
    - Hough, S.E. and Page, M. (2015). A century of induced earthquakes in Oklahoma?. *Bulletin of the Seismological Society of America*, **105**(6), 2863-2870.
    - Kerr, R.A. and Stone, R. (2009). A human trigger for the great quake of Sichuan?. *Science*, **323**(5912), 322-322.
- Klose, C. D. (2008). The 2008 M 7.9 Wenchuan earthquake—Result of local and abnormal mass imbalances?, *Eos Trans. AGU 89*, no. 53, Fall Meet. Suppl., Abstract U21C-08.

We also note that we are currently in the final stages of completing another manuscript exclusively focused on examining the role of anthropogenic processes in the context of natural hazard interactions, which will cover the topic of anthropogenic processes and their influences on natural hazards in much greater detail.

20

10

2. **[Define better anthropogenic processes vs. routine hazard events of technology]** In Section 3.3, the authors state that "similarities exist between the routine hazard events of technology and the definition of anthropogenic processes". Frankly speaking, I do not clearly understand the exact differentiation between both types of hazards from the present discussion. Providing some explicit examples highlighting this difference

25 could be helpful to clarify this point.

Authors: We agree with the reviewer that from the current definitions, these two groups (routine events of technology, anthropogenic processes) are difficult to differentiate. In Section 3.3 we seek to communicate to the reader that there are currently few consistent definitions of technological hazards/disasters, with some

- 30 definitions incorporating anthropogenic processes and others not doing so. We propose that we revise page 6 of our original manuscript, lines 20–22, to clarify that in this definition of technological hazards, the 'routine hazard events of technology' are the same as anthropogenic processes. The explicit example of subsurface mining will be introduced and discussed.
- 35 3. [Unidirectional vs. bi-directional interaction relationships] Regarding their discussion of interaction relationships, I have got the impression that the authors consider such relationships as exclusively unidirectional. Or put differently: the present framework is developed and illustrated with the (in my opinion, unnecessary) implicit restriction of unambiguous cause-effect relationships among hazards (as manifested in the terminology of primary versus secondary hazards). In this regard, I was wondering if there exist any 40 examples of bidirectional interactions, either associated with their simultaneous occurrence or relating to
  - feedbacks. I would be keen to learn about the authors' opinion on this question.

Authors: Thank you for highlighting this confusion in our language, as we do not wish to convey exclusively unidirectional interactions. In order to address the suggestions of the reviewer, we will revise the initial

45 paragraph of **Section 4** (page 7), to explicitly state that interactions can be bidirectional, bringing in some of the discussion below and using an example from Nepal (Marston *et al.*, 1996). In our manuscript we have used the term 'interaction' as it conveys that the relationship between natural hazards could be bidirectional. For example, we acknowledge that feedback mechanisms can be established, where the triggering of a secondary hazard exacerbates the primary hazard, therefore triggering further episodes of the secondary hazard. The

language of primary and secondary hazard was used in Gill and Malamud (2014) and brought into this manuscript. While recognising that this could imply a unidirectional relationship, we believe that as the same 21 natural hazards are used as both primary and secondary hazards it can also be used to discuss bidirectional relationships (including feedback mechanisms), where a triggered secondary hazard exacerbates the primary

- 5 hazard. This is visible in the example used in **Figure 4**, showing a storm triggering a flood, which then triggers a landslide, which exacerbates the flooding. Another specific example of such feedback mechanisms, cited in Gill and Malamud (2014), gives an example from Nepal and discusses the undercutting of slopes by river systems causing channel aggradation. This aggradation can trigger greater undercutting, thus developing a positive feedback or cyclic triggering (Marston *et al.*, 1996).
- 10
- Marston, R., Kleinman, J., & Miller, M. (1996). Geomorphic and forest cover controls on monsoon flooding, central Nepal Himalaya. *Mountain Research and Development*, 257-264.

4. [Causal relationships] In Section 4.1, the authors describe triggering relationships as "causal" relationships (as opposed to the other two types of interactions among hazards). However, depending on which specific
15 notion of causality is considered, I would argue that the other two types also denote some "causal" (in the sense of directional) relations, yet of different quality than triggering relationships where this relation is "most direct".

**Authors**: We agree with the reviewer, that there is a degree of directional 'causation' in all of the interaction 20 types in **Section 4**, and will revise the language in **Section 4** to take this into account.

[Temporal and spatial scales] The authors emphasize (p.7, 1.28) that triggering interactions can occur between a diverse range of hazards and processes. I think that also the issue of scales (both temporal and spatial) is something important to consider when aiming to characterize triggering relationships. It might be 25 interesting to elaborate further on this aspect.

**Authors**: We agree with the reviewer that triggering interactions can occur between a diverse range of hazards and processes, but also that the issue of spatial and temporal scales is very important. We will therefore revise **Section 4.1** in our manuscript to include the following discussion of spatial and temporal scales, with examples

- 30 of volcanic activity and agricultural practice change used to illustrate the importance of considering spatial and temporal scales. We will structure our manuscript to include the following discussion. Spatial and temporal scales are particularly important when going from a global overview of interactions, to a location-specific assessment of interactions. For example, a volcanic eruption could be considered in terms of each individual eruption (occurring over approximately seconds to hours) or a broader period of activity (occurring over
- 35 approximately months to decades). An anthropogenic process, such as agricultural practice change, could also occur at multiple scales. Agricultural practice change could incorporate both an individual farmer ploughing a new field (at an approximate spatial scale of 0.1–1 km2 and temporal scale of days to weeks) and a societal transition from manual to machine-dominated farming (at an approximate spatial scale of 104–107 km2 and temporal scale of years to centuries). Both of these examples would result in diverse characterisations of the
- 40 possible triggering relationships, depending on the scale of interest.

6. **[Redundancy between figure captions and text]** Throughout Section 4, there is considerable redundancy between figure captions and the main text (e.g., between the last paragraph of Section 4.1 and the caption of Figure 2). I think that this redundancy should be reduced.

45

**Authors**: We understand the reviewer's concern and appreciated that this style of writing adds greater repetition. In the revised manuscript we will carefully examine each figure caption to see if any repetition can be removed in the text, as well as considering advice from the editor. However, we both prefer to use detailed figure captions so that each figure stands alone (particularly for the reader that 'dips' in and out of the paper),

50 and can be interpreted easily. These figures are then comprehensively introduced and discussed in the text so

that they are integrated into the discussion within the manuscript. We believe that this makes is easier for the reader to understand and use the figures.

- 7. [Probabilistic and/or individual event viewpoints] In Section 4.2, the authors state that increased
  5 probability relationships "change the frequency or extent of the secondary hazard or process". When considering hazards in some abstract (probabilistic / return period) sense, I agree with this. However, at the level of individual events, a specific primary event would rather change the proximity (and specific characteristics) of the specific next secondary hazard to come affected by the increased probability relationship. From the present paper, it is not always completely clear if the authors wish to consistently take the
- 10 probabilistic or the individual event viewpoint (not necessarily contradicting each other in general).

**Authors**: This is a very helpful and thoughtful comment, and we thank the reviewer for using this opportunity to raise it, as it is one that the two authors have discussed also at length. In general, in our discussions, we are considering probabilistic viewpoints, where the probabilistic behaviour of a relationship is often inferred from

- 15 many individual events. This approach is used to consider in general how one hazard will influence another. We recognise that the approach required when considering a specific case study location may differ to the approach used when discussing populations of interactions in a general and more regional or globally-relevant way. This is a limitation that we have not explicitly brought out in this manuscript, but think that it would be beneficial to do so. We propose adding a paragraph to **Section 2** (*Single vs. Multi-Hazard*), that discusses the
- 20 challenges of adapting global multi-hazard frameworks for use in regional/local contexts, and also the individual event vs. the probabilistic viewpoint.

8. [Frequent vs. rare interaction relationships] In Section 4.3, the authors contrast interaction relationships that "include large numbers of frequently-occurring interactions" and such "that are considered to be less
25 common". This seems to me a rather arbitrary distinction (i.e., there are no two distinct types, but rather a

- continuum of cases according to the frequency of interactions). I was wondering if the authors have any objective criteria for associating catalysing and impedance relationships to any of the two aforementioned types.
- 30 Authors: We thank the reviewer for noticing our confusing choice of phrasing in Section 4.3 which we propose to change to (i) clarify our meaning to avoid confusion, (ii) illustrate more fully our agreement that there is a wide spectrum (continuum) between the two end-points of frequent vs. rare, (iii) bring in specific examples. In Section 4.3 we had aimed to highlight the differential likelihood of different interactions, but agree that this is best illustrated also through examples of different specific interactions, rather than interaction
- 35 types. We also recognise that choosing two examples relating to triggering relationships is unhelpful, given that the focus of **Section 4.3** is on catalysing and impedance relationships. Within each interaction type we will highlight that there can be found examples of frequently occurring interactions and examples that are less common, with a wide spectrum between these two end points. The associated figure, to which our manuscript comment related (**Figure 3**) does not distinguish between these two end members, it represents possible
- 40 interactions. We therefore propose changing the text of **Section 4.3** (page 9, lines 16–22), to focus on examples of catalysing/impedance relationships, which we believe also fall on a spectrum between those that occur frequently and those that are less common. For example, vegetation removal or the unloading of slopes through construction has been shown to increase the susceptibility of slopes to landslides in the event of a trigger (e.g., earthquake or heavy rain). We will refer back to the example of road construction catalysing landslides during
- 45 the 8 October 2005 Kashmir earthquake, cited in **Section 3.2** (Owen *et al.*, 2008). We will also introduce additional supporting literature:
  - Brenning, A., Schwinn, M., Ruiz-Páez, A. P., and Muenchow, J.: Landslide susceptibility near highways is increased by 1 order of magnitude in the Andes of southern Ecuador, Loja province, *Nat. Hazards Earth Syst. Sci.*, 15, 45-57, doi:10.5194/nhess-15-45-2015, 2015.

• Devkota, K.C., Regmi, A.D., Pourghasemi, H.R., Yoshida, K., Pradhan, B., Ryu, I.C., Dhital, M.R. and Althuwaynee, O.F., 2013. Landslide susceptibility mapping using certainty factor, index of entropy and logistic regression models in GIS and their comparison at Mugling–Narayanghat road section in Nepal Himalaya. *Natural Hazards*, 65(1), pp.135-165.

5

• Montgomery, D. R. (1994). Road surface drainage, channel initiation, and slope instability. *Water Resources Research*, 30(6), 1925-1932.

A short discussion of differential likelihoods of triggering interactions will also be added to **Section 4.1** (page 7), using appropriate examples of triggering events with a high, medium and low likelihood, selected from the analysis of *spatial overlap-temporal likelihood* of natural hazard interactions in Gill and Malamud (2014). The

10 revisions to **Sections 4.1** and **4.3** will emphasise the importance of location specific characteristics in determining the likelihood of any interaction or interaction type, when considering applied multi-hazard approaches.

9. [Two vs. four case studies in Section 5] In Section 5, the authors first refer to four case studies, later to
15 two. The first two cases detailed in Section 5.1 are not used any further in the following text, while the other two cases are exclusively discussed in the visualization context. All four examples are clearly relevant in the context of this work, but the way they are introduced and detailed in different parts of the manuscript appears slightly confusing.

- 20 Authors: We have made an error in the introduction to Section 5, which should read that two examples are discussed in Section 5.1, and two different examples are discussed in Section 5.2. In Section 5.1 we introduce two examples of networks of hazard interactions (cascades) to illustrate the wide variation in spatial and temporal extent, frequency of networks of hazard interactions, and impact of such networks. In Section 5.2 we introduce two different examples to discuss the visualisation of networks of hazard interactions. We had
- 25 originally included all four in both Section 5.1 and 5.2, but felt (i) that there was too much repetition if all four examples were included in both Section 5.1 and 5.2, and (ii) if we reduced the examples to just two, the manuscript would not show the same diversity of examples as is achieved with four different examples, spread across the two sections. To address this comment and clarify the way these case studies are introduced and used, we propose adding an additional paragraph at the start of Section 5 (page 10) which gives a short
- 30 summary of all four case studies, and explains clearly that we will use two to illustrate variations in the spatial and temporal extent of networks of interactions, and two to consider the visualisation of such networks. We will then rename **Section 5.1** to '*Spatial and Temporal Extent of Networks of hazard Interactions (Cascades)*'. The title of **Section 5.2** will remain unchanged. We believe that these revisions to structure and content will go some way to addressing the concerns of the reviewer.
- 35

10. **[English]** Despite not being a native speaker, I was wondering about a few words and kindly ask the authors to cross-check them:

(i) p.3, 1.26: "selected"

(ii) p.9, 1.21: "...processes relevant interactions can be..."

40 (iii) p.10, 1.15: "evaluating networks... is important"

(iv) p.15, 1.27: The meaning of the term "Earth-systems management" is not fully obvious to me.

**Authors**: We thank the reviewer for these helpful suggestions, and will cross check each of them when preparing the revised manuscript.

45 (i) "select" vs. "selected". We agree and will change.

(ii) "Only through the careful assessment of all possible single hazards and *processes can relevant interactions* be identified and assessed." Vs. "Only through the careful assessment of all possible single hazards and *processes relevant interactions can be* identified and assessed." We think the first one is correct, but will revisit when writing the revised manuscript.

(iii) "and finally discuss why we believe evaluating networks of hazard interactions are important" vs. "and finally discuss why we believe evaluating networks of hazard interactions is important". We agree, this should be singular (is).

(iv) We note that 'Earth-systems management' is used in the journal description, and aims and scope, but agreethat it would be helpful to add a definition for this term, relating to human actions and decision making that helps to manage aspects of the Earth system.

**REVIEWER #2 (COMMENTS TO AUTHOR): Warner Marzocchi**

**General Comments:** The paper discusses the importance of modeling the interactions among different hazards, natural and not, to obtain a more realistic risk assessment. The paper is well written (maybe with some unnecessary repetitions), and the topic of great interest. My general opinion is positive but I think that the paper misses to take into account some key issues for a proper multi-hazard assessment.... To sum, I am

15 positive about this paper because it reiterates the importance of the interaction among different hazards in a multi-risk perspective. This is certainly commendable. However, I think that the authors should make an effort to address the points raised [below].

Authors: We are very grateful for this positive, constructive and helpful summary, and appreciate the thoughtful comments you have made below to help us improve the manuscript. We have attempted to address each of your key issues below and will welcome further dialogue. We have added some clarifying words at the beginning of each of your comments in [] to give a highlight of what the point is about.

25 Major Point 1. [Forecasting time windows] The first issue, and probably the most important, is that the authors do not discuss the forecasting time windows for hazard assessment. The importance of this topic for multi-hazard and multi-risk assessment has been already discussed in Marzocchi *et al.* (2012; *W. Marzocchi, A. Garcia-Aristizabal, P. Gasparini, M.L. Mastellone, A. Di Ruocco, 2012. Basic principles of multi-risk assessment: a case study in Italy. Natural Hazards, 62, 551-573*), and it will be summarized here.

The 'hazard' is essentially the probability of a threatening event in one specific time-space window. The (forecasting) time window is of critical importance and it is usually related to the specific use of the hazard assessment in terms of risk reduction. For instance, a long-term hazard assessment (e.g., a forecasting time window of 50 years) is usually adopted for land use planning, like to define the building code for the

- 35 earthquakes threat. On the other hand, the management of rapidly evolving emergencies usually requires short-term assessments (e.g., days to weeks). Considering long- or short-term hazard provides completely different scenarios for hazard interactions. For example, let us consider the landslide hazard. We can calculate the probabilities of landslides occurrence for the next decades just simply looking at the historical catalog. The fact that most of landslides are caused by earthquakes is not relevant in the long-term hazard because this is
- 40 true also for the landslides occurred in the past and reported in the historical catalog (assuming that the longterm earthquake rate is not changing with time). So, the interaction between earthquakes and landslides could be irrelevant for the long-term hazard assessment. Of course, if we are considering the short-term hazard assessment (as I guess the authors are doing), the fact that a large earthquake (like the Gorkha event) has just occurred modifies significantly the probability of landslides in the next few months (due to the occurrence of
- 45 aftershocks). So, the interaction between hazards has to be considered specifying clearly the forecasting time window. Marzocchi *et al.* (2012) explore in detail this point showing a probabilistic framework to explain when hazard interactions are relevant or not (see section 3.2).

Authors: We thank the reviewer for raising the very interesting and relevant point of forecasting time 50 windows. We will add the subject of forecasting windows in Section 2 as an important consideration to our manuscript, including reference to the referee's manuscript, along with two others.

- Marzocchi, W., Garcia-Aristizabal, A., Gasparini, P., Mastellone, M.L., Di Ruocco, A. (2012). Basic principles of multi-risk assessment: a case study in Italy. Natural Hazards, 62, 551-573.
- Kappes, M. S., Keiler, M., von Elverfeldt, K., & Glade, T. (2012). Challenges of analyzing multihazard risk: a review. Natural Hazards, 64(2), 1925-1958.

5

Selva, J. (2013). Long-term multi-risk assessment: statistical treatment of interaction among risks. Natural hazards, 67(2), 701-722.

We will revise our manuscript to add a paragraph to Section 2 that describes our general, globally-relevant approach (generalising across forecasting time windows that are short- and long-term, discussed below), but that acknowledges the importance of establishing clear boundary conditions when developing location specific 10 multi-hazard risk/multi-risk assessments.

We agree that the forecasting time window is important and that this will differ depending on the use of a particular multi-hazard risk assessment (MHRA)/multi-risk assessment (MRA) [N.B., please see our response below to Major Comment 2, regarding use of MHRA vs. MRA]. We agree with Marzocchi et al. (2012) that

- 15 prior to developing a MHRA/MRA the space-time assessment window should be clearly stated, and will be determined based on the requirements of the users of the MHRA/MRA. We will state this more clearly in our revision. We clarify here that in the context of this manuscript we are not focusing on a specific application of a MRA or a specific forecasting time window, rather we are generalising across different forecasting time windows in order to describe what may occur. With reference to whether or not interactions are relevant, from
- 20 the perspective we have taken in explaining interactions, we believe that the interactions between certain hazards are relevant for long-term hazard assessments. Marzocchi et al. (2012), for example, notes that tsunami databases already include the possibility that a tsunami is caused by an earthquake. Therefore, our understanding of your argument is that the interaction 'earthquake  $\rightarrow$  tsunami' is irrelevant in the long-term hazard assessment of tsunamis (i.e., a tsunami database already takes into account the preceding earthquake).
- 25 We do not dispute that the database of tsunamis already takes into account the tsunami triggers, and note that it is an important point. However, even in this long-term forecasting time window it is important to acknowledge this interaction (even if only that the interaction can occur) to ensure clear and coherent communication on the spatial/temporal association of tsunamis with earthquakes, to all stakeholders. We do not dispute the importance that you are stating, and illustrated in Marzocchi et al. (2012), but believe in the
- 30 context of our argument of potential hazard interactions, both long-term and short-term forecasting are included (i.e., we are not considering 'just' short-term time windows).

If you would like to correspond further on this issue, either through the discussion journal platform, or via email, we would be happy to do this.

35

Major Point 2. [Hazard vs. risk] The second issue is that the authors seem to confuse the hazard with the risk. Hazard is an essential component of risk assessment, but it is not the risk. So, I think that introducing vulnerability interactions in the multi-hazard assessment is not proper (like they did in their figure 1). Again, Marzocchi et al. (2012) addressed this distinction in their Figure 2. I emphasize that this distinction is not only

40 semantic. In a multi-hazard/risk perspective, it is important to mention that hazards cannot be (usually) compared, while the risks can. In fact, hazards are usually represented by curves that represent the exceedance probability of some quantity of interest, like ash fall thickness, ground shaking acceleration, etc. So it is not clear to me how we can compare them meaningfully and say when a hazard is higher than another one. On the other hand, the associated risks can be easily compared.

45

Authors: We agree with the reviewer that there is a substantive difference between hazard and risk, and do not wish to convey in our manuscript that these are interchangeable. We propose revising the figure caption for Figure 1 and associated text in Section 2 (pages 3-4) to make our meaning clearer, and remove any confusion about where we are referring to hazard and where we are referring to risk. Our Figure 1 (from Gill and Malamud, 2014) originally related to the transition from multi-layer single hazard *risk* assessments and multi-hazard *risk* assessments. Our reading of the multi-hazard literature suggested that it was common to use the term 'multi-hazard risk', although we also acknowledge that multi-risk is a valid alternative.

- **5 Major Point 3. [Triggered vs. increased probability]** The final issue is about the distinction between "triggered" and "increase probability" made by the authors. This distinction is not clear to me. It seems that the authors assume that triggering is a deterministic link between hazards. In my view the concept of triggering and of increase probability are identical, at least this is true in seismic and volcanic hazard. We can talk of deterministic causal relationship only retrospectively (when you have already observed that one event triggered
- 10 another one). But when we see the problem in a prospective way, the occurrence of an event \*may\* increase the probability to trigger another event, but very rarely we can be sure about this triggering. Of course the authors can mean something different, but I think that this distinction has to be clarified.

Authors: This is a thoughtful and stimulating comment, and we appreciate the reviewer's contribution. We propose, following the below outline of ideas, adding a discussion in **Section 4** of our manuscript, where we outline our reasons for distinguishing triggered and increased probability as two different groups, while also acknowledging these two terms (triggering, increase probability) are end members. We will also will highlight that we are not 'only' discussing deterministic links, and that randomness (stochasticity) is certainty a key element. We believe the manuscript will benefit from a stronger explanation.

20

50

We propose two themes where one can differentiate between one event *triggering* another event, and one event *increasing the probability* of another event.

- Theme 1. Direct vs. indirect sequence of physical processes between the primary and secondary 25 hazard. Although we recognize there can be a continuum between 'triggered' and 'increased probability' relationships in terms of meaning, we believe that one can roughly (and at a certain level, intuitively) differentiate between these relationships by considering direct vs. indirect sequences of interacting physical processes between the primary and secondary hazards. For an example of a (roughly) direct sequence, the addition of water to geological material on a hillslope can directly 30 trigger landslides (heavy rain  $\rightarrow$  landslides). In contrast, an example (roughly) of an indirect sequence, is a forest fire removing vegetation which in turn reduces the shear strength of the slope and is thus much more prone to landsliding in the future if there is an earthquake or heavy rain (forest fire  $\rightarrow$ reduced slope strength  $\rightarrow$  [with external trigger of an earthquake] landslides). In this example, some of the population of landslides may occur during a rain or earthquake event that would not have 35 occurred if the forest fire had not occurred. Another example of an indirect sequence is the impact of ground subsidence on flooding. Subsidence in itself may not trigger a flood, however it could make flooding more likely to occur in the event of a river spilling over its banks, a storm surge or a tsunami.
- *Theme 2. Temporal sequences.* We believe it is also possible to differentiate between triggering and increased probability relationships by considering the timing of the sequence of events. Take the following time sequence (arbitrary units and lengths of time for the windows):

**Time Window 1 [Primary Hazard Window] Time Window 2A Time Window 2B**

- 45 Using this time sequence, both forward looking (i.e., during **Time Window 1** which occurs before the primary hazard) and retrospective approaches (i.e., during **Time Window 2A** and **2B** which occurs after the primary hazard, and includes the secondary hazard) can help differentiate between triggering and increased probability relationships. As an illustrative example, we will take an earthquake as the primary hazard, and a triggered landslide population event as the secondary hazard:
  - While in Time 1 what will happen in Time 2A and 2B [forward looking]? Prior to an earthquake [*primary hazard*] occurring [Time 1], it can be stated that given an earthquake, landslides [*secondary hazard*] may occur in the time period after the earthquake (e.g., minutes

to days) [**Time 2A**]. Furthermore, while in **Time 1**, we can state that more landslides may occur much later after an earthquake event (months to years) [Time 2B] due to reduced shear strength of the slope (similar to an indirect sequence of physical processes given in Theme 1 above). A different response is likely to be needed at an operational level for these two timings.

- 5 While in the interface between Time 2A and 2B, what has occurred in Time 1 and 2A [retrospective] and Time 2B [forward looking]? Immediately after an earthquake [primary hazard] has occurred, we can identify any landslides [secondary hazard] that were triggered by the earthquake. We can also look forward to **Time 2B**, and state that there is now an increased probability of landslides due to the decreased shear strength of the slopes in the region.
  - While at end of Time 2B, look back at Time 1, 2A and 2B [retrospective]. At the end of **Time 2B**, we look back retrospectively to assess what landslides have been triggered directly by the earthquake (done through triggered landslide event inventories) and those landslides that have resulted from decreased shear strength (more difficult to quantify in actual practice).
- 15 When generalising across these three time windows, it is perhaps helpful therefore to recognise that an earthquake [primary hazard] can both trigger and increase the likelihood of landslides [secondary *hazard*] occurring in [Time 2A and 2B].

Therefore, while we agree that there are similarities between these two interactions types in that they both 20 represent a change in probability of a secondary hazard (e.g., landslide) given a primary hazard (e.g., earthquake), we would suggest that they can be characterised by two end-member types (with a continuum between them):

- i. *Triggering:* A probability associated with a threshold being reached or passed.
- ii. Increased Probability: A probability associated with a change in environmental parameters so as to 25 move towards, but not reach a particular threshold.

We would suggest that for the purposes of operational hazard management, it is beneficial to consider both (i) and (ii) above, distinguishing between the two interaction types. For any given window of time after a primary hazard, agencies such as civil protection professionals may want to know what the likelihood is of landslides occurring (being *triggered*), but also whether there is a change in the likelihood of landslides beyond this

30 window of time (increased probability). Again, if you would like to correspond further on this issue, either through the discussion journal platform, or via e-mail, we would be happy to do this.

Optional Point 1. As a final suggestion, I would like to see a little bit more emphasis on the probabilistic and quantitative nature of hazard assessment. This is just a suggestion and it is not mandatory, but I do think that

35 quantitative assessment of the hazard (and of the consequent risk) is the only possible strategy to plan rational risk reduction actions.

Authors: We agree with this reviewer (and the first reviewer) that there is an important role for quantitative hazard assessment methods. In our revised manuscript will provide a more substantive discussion of 40 quantitative hazard assessment, notably in Section 2 (Single vs. Multi-Hazard).

10

---

## Author Response (AR1)

**Response to Reviewer Comments (Based on Revised Manuscript)**

**Title: Hazard Interactions and Interaction Networks (Cascades) within Multi-Hazard Methodologies**

**[doi:10.5194/esd-2015-94] submitted to *Earth System Dynamics***

We thank **Reviewer #1 (Reik Donner)** and **Reviewer #2 (Warner Marzocchi)** for their positive, insightful and constructive comments on our manuscript, and for waiving their right to anonymity. We also appreciate the guidance and additional time granted to us by the editor, Christian Franzke, to complete revisions to this manuscript. We believe that the comments made by both reviewers were fair and have improved our overall structure and argument, and thank both **Reik Donner** and **Warner Marzocchi** for their time and assistance.

Following the initial responses that we submitted to reviewers' comments on 14 March 2016 (uploaded to the discussion journal), we have revised our manuscript. Here we present our final response to each of the reviewers' comments and explain how we have changed our manuscript to reflect these comments. We also attach a track change document showing changes between this version of the manuscript and the initial one submitted to the journal. The main revisions we have made are in response to reviewer comments, with changes including some deletion of text, addition and reordering of text for clarity, and sub-dividing of some headers into sub-headers for increased coherence. No figures have had major changes (only minor corrections) and no figures have been added. Some changes have been made to figure captions. We have added 18 references and deleted 3 references.

In addition to the reviewer comments, we have taken this opportunity to reread the text and make some minor corrections and clarifications in the text and references (also shown using track changes).

We hope that this final revision is satisfactory, and look forward, if accepted, to dealing with any final copy-editing that might be necessary.

We look forward to your comments.

Joel C. Gill and Bruce D. Malamud

(Department of Geography, King's College London, UK, corresponding author: joel.gill@kcl.ac.uk)

**REVIEWER #1 (COMMENTS TO AUTHOR): Reik Donner [RD]**

**RD General comments:** Gill and Malamud discuss a general framework for extending the commonly taken view on hazard assessment by taking interdependences of different types linking various hazards into account. Since such interdependences may significantly affect the susceptibility of regions to hazards of different types, I fully agree with their basic argument that hazard interactions need to be carefully considered for obtaining qualitative and quantitative risk assessments. The discussion presented by the authors is scientifically sound and does not only highlight the insufficiencies of previous approaches, but also proposes possible ways to reaching such realistic assessment. The latter aspect appears to be the main achievement of the present work that goes beyond what has been discussed by the authors in their recent review paper (Gill and Malamud, 2014). At the present stage, this work is purely qualitative, and future studies describing its further extension towards quantitative multi-hazard risk assessments would be most welcome. To this end, this excellent work can be considered as an important milestone on this way. While the manuscript already presents excellent work, I would like to recommend additionally paying specific attention to some minor points listed below to possibly make it even better.

**Authors:** We are very grateful for this positive, constructive and helpful summary, recognising the value of this contribution to the literature on multi-hazard assessments. We have attempted to address each of the points that

you have listed below, and appreciate your thoughtful interaction with our manuscript. We have added some clarifying words at the beginning of each of your comments in **[ ]** to give a highlight of what the point is about.

**RD** 1. **[Induced seismicity as an example of anthropogenic processes]** Regarding the anthropogenic processes briefly discussed in Section 3.2, I think that another possibly important example of such processes would be induced seismicity, i.e., seismic activity resulting from human activities like construction of large artificial reservoirs or mining/drilling activities. Recent corresponding references on possible cases of induced seismicity include, among others, Kerr and Stone, Science 323, 322, 2009, or Hough and Page, *Bulletin of the Seismological Society of America*, doi: 10.1785/0120150109. I would like to suggest adding some brief comment on such anthropogenic processes.

**Authors**: We agree that induced seismicity is an important example of anthropogenic processes resulting in the triggering of a natural hazard, in this case seismic activity. Following the helpful suggestion, we have included induced seismicity in a revised version (addition of two sentences) of **Section 3.2**, introducing the following additional papers to the manuscript:

- Ellsworth, W. L.: Injection-induced earthquakes. *Science* 341(6142), 142-149, doi:10.1126/science.1225942, 2013.
- González, P. J., Tiampo, K. F., Palano, M., Cannavó, F., & Fernández, J. (2012). The 2011 Lorca earthquake slip distribution controlled by groundwater crustal unloading. *Nature Geoscience*, **5**(11), 821-825.
- Hough, S.E. and Page, M. (2015). A century of induced earthquakes in Oklahoma?. *Bulletin of the Seismological Society of America*, 105(6), 2863-2870.
- Simpson, D. W.: Seismicity changes associated with reservoir loading, *Eng. Geol.*, 10(2), 123-150, 1976.

We have also added some of the causes of induced seismicity (reservoir construction, wastewater injection) into **Table 1**. We note that we are currently in the final stages of completing another manuscript exclusively focused on examining the role of anthropogenic processes in the context of natural hazard interactions, which will cover the topic of anthropogenic processes and their influences on natural hazards in much greater detail.

**RD** 2. **[Define better anthropogenic processes vs. routine hazard events of technology]** In Section 3.3, the authors state that "similarities exist between the routine hazard events of technology and the definition of anthropogenic processes". Frankly speaking, I do not clearly understand the exact differentiation between both types of hazards from the present discussion. Providing some explicit examples highlighting this difference could be helpful to clarify this point.

**Authors**: We agree with the reviewer that from the current definitions, these two groups (routine hazard events of technology, anthropogenic processes) are difficult to differentiate. In **Section 3.3** we seek to communicate to the reader that there are currently few consistent definitions of technological hazards/disasters, with some definitions incorporating anthropogenic processes and others not doing so. We have therefore revised **Section 3.3**, to clarify that in this definition of technological hazards, the 'routine hazard events of technology' are the same as anthropogenic processes. The explicit example of surface mining is also introduced and discussed in the revised **Section 3.3**, discussing how this example can be considered as both an anthropogenic process and a routine hazard event of technology. We believe that our discussion of the overlap that exists between these two groups (routine hazard events of technology, anthropogenic processes) is now communicated more clearly.

**RD** 3. **[Unidirectional vs. bi-directional interaction relationships]** Regarding their discussion of interaction relationships, I have got the impression that the authors consider such relationships as exclusively unidirectional. Or put differently: the present framework is developed and illustrated with the (in my opinion, unnecessary) implicit restriction of unambiguous cause-effect relationships among hazards (as manifested in the terminology of primary versus secondary hazards). In this regard, I was wondering if there exist any examples of bidirectional

interactions, either associated with their simultaneous occurrence or relating to feedbacks. I would be keen to learn about the authors' opinion on this question.

**Authors**: Thank you for highlighting this confusion in our language, as we do not wish to convey exclusively unidirectional interactions. In order to address the suggestions of the reviewer, we have added an extensive new set of text in our revised manuscript **Section 4.1** to explicitly state that interactions can be either unidirectional or bidirectional, bringing in some of the discussion below and using theoretical examples to illustrate both of these relationships. In our manuscript we have used the term 'interaction' as it communicates the potential for unidirectional and bidirectional relationships. The language of primary and secondary hazard was used in Gill and Malamud (2014) and brought into this manuscript. While recognising that this could imply a unidirectional relationship, we believe that as the same 21 natural hazards are used as both primary and secondary hazards it can also be used to discuss bidirectional relationships (including feedback mechanisms), where a triggered secondary hazard exacerbates the primary hazard. Additional examples discussed in the manuscript include:

- [Unidirectional] First the 'primary' hazard occurs and then the 'secondary' hazard. An example is a tropical storm triggering a flood. In this case the flood may trigger further hazards (e.g., ground collapse, ground heave), but there is no feedback from the flood back to the tropical storm.
- [Bidirectional] Feedback mechanisms may occur where a primary hazard triggers a secondary hazard which exacerbates the primary hazard, therefore triggering further episodes of the secondary hazard. An example of this would be a landslide blocking a river, resulting in a flood, but then the water upstream of the blockage interacting with the original landslide, breaking it down, and the water potentially triggering further landslides.

**RD** 4. **[Causal relationships]** In Section 4.1 *[Section 4.2 in revised manuscript]*, the authors describe triggering relationships as "causal" relationships (as opposed to the other two types of interactions among hazards). However, depending on which specific notion of causality is considered, I would argue that the other two types also denote some "causal" (in the sense of directional) relations, yet of different quality than triggering relationships where this relation is "most direct".

**Authors**: We agree with the reviewer, that there is a degree of directional 'causation' in all of the interaction types in **Section 4**, and have revised the language in the revised manuscript **Section 4.2** (in addition to other appropriate places in the manuscript) to take this into account.

**RD** 5. **[Temporal and spatial scales]** The authors emphasize (p.7, l.28) that triggering interactions can occur between a diverse range of hazards and processes. I think that also the issue of scales (both temporal and spatial) is something important to consider when aiming to characterize triggering relationships. It might be interesting to elaborate further on this aspect.

**Authors**: We agree with the reviewer that triggering interactions can occur between a diverse range of hazards and processes, but also that the issue of spatial and temporal scales is very important. In **Section 4.2** of our revised manuscript we now include a discussion of spatial and temporal scales, using the example of agricultural practice change to illustrate their importance. Agricultural practice change could incorporate both an individual farmer ploughing a new field (at an approximate spatial scale of $0.1$–$1$ km$^2$ and temporal scale of days to weeks) and a societal transition from manual to machine-dominated farming (at an approximate spatial scale of $10^4$–$10^7$ km$^2$ and temporal scale of years to centuries). Differences in the scale of interest of agricultural practice change would result in diverse characterisations of the possible triggering relationships.

**RD** 6. **[Redundancy between figure captions and text]** Throughout Section 4, there is considerable redundancy between figure captions and the main text (e.g., between the last paragraph of Section 4.1 *[Section 4.2 in revised manuscript]* and the caption of Figure 2). I think that this redundancy should be reduced.

**Authors**: We understand the reviewer's concern and appreciated that this style of writing adds greater repetition. In the revised manuscript we have examined each figure caption to see if any repetition can be removed in the text. However, we both prefer to use detailed figure captions so that each figure stands alone (particularly for the reader that 'dips' in and out of the paper), and can be interpreted easily. These figures are then comprehensively introduced and discussed in the text so that they are integrated into the discussion within the manuscript. We believe that this makes is easier for the reader to understand and use the figures.

**RD** 7. **[Probabilistic and/or individual event viewpoints]** In Section 4.2 *[Section 4.3 in revised manuscript]*, the authors state that increased probability relationships "change the frequency or extent of the secondary hazard or process". When considering hazards in some abstract (probabilistic / return period) sense, I agree with this. However, at the level of individual events, a specific primary event would rather change the proximity (and specific characteristics) of the specific next secondary hazard to come affected by the increased probability relationship. From the present paper, it is not always completely clear if the authors wish to consistently take the probabilistic or the individual event viewpoint (not necessarily contradicting each other in general).

**Authors**: This is a very helpful and thoughtful comment, and we thank the reviewer for using this opportunity to raise it, as it is one that we have discussed also at length between ourselves. In general, in our manuscript, we are considering probabilistic viewpoints, where the probabilistic behaviour of a relationship is often inferred from many individual events. This approach is used to consider in general how one hazard will influence another. We recognise that the approach required when considering a specific case study location may differ to the approach used when discussing populations of interactions in a general and more regional or globally-relevant way. This is a limitation that we had not explicitly brought out in the original manuscript, but have integrated into the revised manuscript. We have added a new manuscript section (**Section 2.2**) (*From Global to Local Multi-Hazard Approaches*), that discusses the challenges of adapting global multi-hazard frameworks for use in regional/local contexts, and also the individual event vs. the probabilistic viewpoint. We also discuss this point (together with other comments raised by both reviewers) in our revised manuscript **Section 4.1**, Classifying Interaction Types.

**RD** 8. **[Frequent vs. rare interaction relationships]** In Section 4.3 *[Section 4.4 in revised manuscript]*, the authors contrast interaction relationships that "include large numbers of frequently-occurring interactions" and such "that are considered to be less common". This seems to me a rather arbitrary distinction (i.e., there are no two distinct types, but rather a continuum of cases according to the frequency of interactions). I was wondering if the authors have any objective criteria for associating catalysing and impedance relationships to any of the two aforementioned types.

**Authors**: We thank the reviewer for noticing our confusing choice of phrasing in our original **Section 4.3** (revised manuscript **Section 4.4**) which we have revised to (i) clarify our meaning to avoid confusion, (ii) illustrate more fully our agreement that there is a wide spectrum (continuum) between the two end-points of frequent vs. rare, (iii) bring in specific examples. In our initial submission, we had aimed to highlight the differential likelihood of different interactions, but agree that this is best illustrated also through examples of different specific interactions, rather than interaction types. We also recognise that choosing two examples relating to triggering relationships was (in our original submission) unhelpful, given that the focus of this section is on catalysing and impedance relationships. Within each interaction type we have now highlighted that there can be found examples of frequently occurring interactions and examples that are less common, with a wide spectrum between these two end-members. The associated figure, to which our manuscript comment related (**Figure 3**) does not distinguish between these two end-members, it represents possible interactions. We have therefore changed the text of our revised manuscript **Section 4.4**, to focus on examples of catalysing/impedance relationships, which we believe also fall on a spectrum between those that occur frequently and those that are less common. For example, vegetation removal or the unloading of slopes through construction has been shown to increase the susceptibility of slopes to landslides in the event of a trigger (e.g., earthquake or heavy rain). We have referred back to the

example of road construction catalysing landslides during the 8 October 2005 Kashmir earthquake, cited in **Section 3.2** (Owen *et al.,* 2008), and introduced additional supporting literature:

- Brenning, A., Schwinn, M., Ruiz-Páez, A. P., and Muenchow, J.: Landslide susceptibility near highways is increased by 1 order of magnitude in the Andes of southern Ecuador, Loja province, *Nat. Hazards Earth Syst. Sci*., 15, 45-57, doi:10.5194/nhess-15-45-2015, 2015.
- Devkota, K.C., Regmi, A.D., Pourghasemi, H.R., Yoshida, K., Pradhan, B., Ryu, I.C., Dhital, M.R. and Althuwaynee, O.F., 2013. Landslide susceptibility mapping using certainty factor, index of entropy and logistic regression models in GIS and their comparison at Mugling–Narayanghat road section in Nepal Himalaya. *Natural Hazards*, 65(1), pp.135-165.
- Montgomery, D. R. (1994). Road surface drainage, channel initiation, and slope instability. *Water Resources Research*, 30(6), 1925-1932.

A short discussion of differential likelihoods of triggering interactions has also been added to our revised manuscript **Section 4.2**, introducing an analysis of *spatial overlap-temporal likelihood* of triggering relationships from Gill and Malamud (2014). The modifications to the revised manuscript **Sections 4.2** and **4.4** now emphasise the importance of location specific characteristics in determining the likelihood of any interaction or interaction type, when considering applied multi-hazard approaches.

**RD** 9. **[Two vs. four case studies in Section 5]** In Section 5, the authors first refer to four case studies, later to two. The first two cases detailed in Section 5.1 are not used any further in the following text, while the other two cases are exclusively discussed in the visualization context. All four examples are clearly relevant in the context of this work, but the way they are introduced and detailed in different parts of the manuscript appears slightly confusing.

**Authors**: We agree with the reviewer that the way in which case studies were introduced and used in **Section 5** was confusing, and have redrafted and re-ordered **Section 5** in the revised manuscript to form a clearer argument. In the revised manuscript **Section 5.1**, we described relevant hazards and processes in Guatemala, and summarise three case studies of networks of interacting hazards (cascades) from Guatemala, with one additional example from Nepal. In the revised manuscript **Section 5.2**, we now discuss variations in spatial and temporal extent, frequency and impact of networks of hazard interactions (cascades), referring back to some of these examples. In the revised manuscript **Section 5.3** we discuss the visualisation frameworks that can be used to represent cascades, and use case studies from revised manuscript **Section 5.1** and other theoretical examples. In the revised manuscript **Section 5.4**, we then discuss the importance of networks of hazard interactions (cascades). We believe that this restructuring, together with enhanced 'signposting' of how each case study will be used, has made **Section 5** much more accessible to the reader.

**RD** 10. **[English]** Despite not being a native speaker, I was wondering about a few words and kindly ask the authors to cross-check them:
(i) p.3, l.26: "selected"
(ii) p.9, l.21: "...processes relevant interactions can be..."
(iii) p.10, l.15: "evaluating networks... is important"
(iv) p.15, l.27: The meaning of the term "Earth-systems management" is not fully obvious to me.

**Authors**: We thank the reviewer for these helpful suggestions, and have addressed each in the revised manuscript as follows:
(i) "select" vs. "selected". We agree and have made the change.
(ii) "Only through the careful assessment of all possible single hazards and *processes can relevant interaction*s be identified and assessed." Vs. "Only through the careful assessment of all possible single hazards and *processes relevant interactions can be* identified and assessed." We think the first one was correct, and have retained this phrasing.

(iii) "and finally discuss why we believe evaluating networks of hazard interactions are important" vs. "and finally discuss why we believe evaluating networks of hazard interactions is important". We agree, this should be singular (is) and have made the change.

(iv) We note that 'Earth-systems management' is used in the journal description, and aims and scope, but agree that it would be helpful to better define this in the context of the sentence. We have used the alternative sentence "*improve management of those aspects of the Earth system that are relevant to disaster risk reduction*" in a revised **Section 6**.

**REVIEWER #2 (COMMENTS TO AUTHOR): Warner Marzocchi (WM)**

**WM General Comments:** The paper discusses the importance of modeling the interactions among different hazards, natural and not, to obtain a more realistic risk assessment. The paper is well written (maybe with some unnecessary repetitions), and the topic of great interest. My general opinion is positive but I think that the paper misses to take into account some key issues for a proper multi-hazard assessment.… To sum, I am positive about this paper because it reiterates the importance of the interaction among different hazards in a multi-risk perspective. This is certainly commendable. However, I think that the authors should make an effort to address the points raised [below].

**Authors:** We are very grateful for this positive, constructive and helpful summary, and appreciate the thoughtful comments you have made below to help us improve the manuscript. We have attempted to address each of your key issues below. We have added some clarifying words at the beginning of each of your comments in **[ ]** to give a highlight of what the point is about.

**WM Major Point 1**. **[Forecasting time windows]** The first issue, and probably the most important, is that the authors do not discuss the forecasting time windows for hazard assessment. The importance of this topic for multi-hazard and multi-risk assessment has been already discussed in Marzocchi *et al.* (2012*; W. Marzocchi, A. Garcia-Aristizabal, P. Gasparini, M.L. Mastellone, A. Di Ruocco, 2012. Basic principles of multi-risk assessment: a case study in Italy. Natural Hazards, 62, 551-573*), and it will be summarized here.

The 'hazard' is essentially the probability of a threatening event in one specific time-space window. The (forecasting) time window is of critical importance and it is usually related to the specific use of the hazard assessment in terms of risk reduction. For instance, a long-term hazard assessment (e.g., a forecasting time window of 50 years) is usually adopted for land use planning, like to define the building code for the earthquakes threat. On the other hand, the management of rapidly evolving emergencies usually requires short-term assessments (e.g., days to weeks). Considering long- or short-term hazard provides completely different scenarios for hazard interactions. For example, let us consider the landslide hazard. We can calculate the probabilities of landslides occurrence for the next decades just simply looking at the historical catalog. The fact that most of landslides are caused by earthquakes is not relevant in the long-term hazard because this is true also for the landslides occurred in the past and reported in the historical catalog (assuming that the long-term earthquake rate is not changing with time). So, the interaction between earthquakes and landslides could be irrelevant for the long-term hazard assessment. Of course, if we are considering the short-term hazard assessment (as I guess the authors are doing), the fact that a large earthquake (like the Gorkha event) has just occurred modifies significantly the probability of landslides in the next few months (due to the occurrence of aftershocks). So, the interaction between hazards has to be considered specifying clearly the forecasting time window. Marzocchi *et al.* (2012) explore in detail this point showing a probabilistic framework to explain when hazard interactions are relevant or not (see section 3.2).

**Authors**: We thank the reviewer for raising the very interesting and relevant point of forecasting time windows. We also thank him for the further communications we had via e-mail after we posted our initial reply (on-line). As a result, we have added a new section within **Section 2** (**Section 2.2**, *From Global to Local Multi-Hazard*

*Approaches*) that discusses forecasting time windows, including reference to the reviewer's manuscript, along with two others.

- Marzocchi, W., Garcia-Aristizabal, A., Gasparini, P., Mastellone, M.L., Di Ruocco, A. (2012). Basic principles of multi-risk assessment: a case study in Italy. *Natural Hazards*, 62, 551-573.
- Kappes, M. S., Keiler, M., von Elverfeldt, K., & Glade, T. (2012). Challenges of analyzing multi-hazard risk: a review. *Natural Hazards*, **64**(2), 1925-1958.
- Selva, J. (2013). Long-term multi-risk assessment: statistical treatment of interaction among risks. *Natural hazards*, **67**(2), 701-722.

We have also used this idea to address WM Major Point 3 (see below) with a detailed addition of text in our revised manuscript **Section 4.1**, where we discuss explicitly different time windows in relation to primary-secondary hazard interactions. In the new **Section 2.2** we describe the contrasts between our general, globally-relevant approach (generalising across forecasting time windows that are short- and long-term, discussed below), and also acknowledge the importance of establishing clear temporal limits when developing location specific multi-hazard risk/multi-risk assessments. We agree that the forecasting time window is important and that this will differ depending on the length of time considered and the use of a particular multi-hazard risk assessment (MHRA)/multi-risk assessment (MRA) [N.B., please also see our response below to WM Major Point 2, regarding use of MHRA vs. MRA]. In the context of our manuscript we are not focusing on a specific application of a MRA or a specific forecasting time window, rather we are generalising across different forecasting time windows in order to describe what *may* occur. We agree with the reviewer that when constructing location-specific assessments of hazard potential, hazard interactions are of greater importance when assessing short-term hazard (e.g., just after an earthquake has occurred, that the probability of landslides has increased), and of less importance when considering long-term hazard (e.g., considering the long-term record of earthquakes from a catalogue). However, as we are also considering interactions for the purposes of hazard education, communication and disaster risk reduction (i.e., not just hazard assessment), it is important to consider interactions in both short and long-term contexts to aid the understanding of natural hazards. We do not believe this to be in conflict with what is included in Marzocchi *et al.* (2012) or outlined in Major Point 1, and hope that our revised manuscript **Section 2.2** and **Section 4.1** (see below for more detailed explanation) now goes, at least part way, towards addressing this issue.

**WM Major Point 2. [Hazard vs. risk]** The second issue is that the authors seem to confuse the hazard with the risk. Hazard is an essential component of risk assessment, but it is not the risk. So, I think that introducing vulnerability interactions in the multi-hazard assessment is not proper (like they did in their figure 1). Again, Marzocchi *et al.* (2012) addressed this distinction in their Figure 2. I emphasize that this distinction is not only semantic. In a multi-hazard/risk perspective, it is important to mention that hazards cannot be (usually) compared, while the risks can. In fact, hazards are usually represented by curves that represent the exceedance probability of some quantity of interest, like ash fall thickness, ground shaking acceleration, etc. So it is not clear to me how we can compare them meaningfully and say when a hazard is higher than another one. On the other hand, the associated risks can be easily compared.

**Authors**: We agree with the reviewer that there is a substantive difference between hazard and risk, and do not wish to convey in our manuscript that these are interchangeable. We have revised the figure caption for **Figure 1** and associated text in the revised manuscript **Section 2.1** to make our meaning clearer, and remove any confusion about where we are referring to hazard and where we are referring to risk (and include statements that 'risk' includes hazard, exposure and vulnerability, but that our paper focusses for the most part on the hazard component). Our **Figure 1** (from Gill and Malamud, 2014) originally related to the transition from multi-layer single hazard assessments to multi-hazard *risk* assessments (where hazard interactions, hazard coincidence and vulnerability [exposure] are included). Our reading of the multi-hazard literature suggested that it is common to use the term 'multi-hazard risk', although we also acknowledge that multi-risk is a valid alternative.

**WM Major Point 3. [Triggered vs. increased probability]** The final issue is about the distinction between "triggered" and "increase probability" made by the authors. This distinction is not clear to me. It seems that the authors assume that triggering is a deterministic link between hazards. In my view the concept of triggering and of increase probability are identical, at least this is true in seismic and volcanic hazard. We can talk of
5 deterministic causal relationship only retrospectively (when you have already observed that one event triggered another one). But when we see the problem in a prospective way, the occurrence of an event \*may\* increase the probability to trigger another event, but very rarely we can be sure about this triggering. Of course the authors can mean something different, but I think that this distinction has to be clarified.

10 **Authors**: This is a thoughtful and stimulating comment, and we appreciate the reviewer's contribution. We have addressed this comment in detail in our revised manuscript **Section 4.1**, giving a justification of our decision to distinguish triggered and increased probability as two different groups, while also acknowledging that these two terms (triggering, increase probability) can be considered end-members. To aid our detailed discussion, we bring into our revised manuscript, the following time windows surrounding the primary window (arbitrary units and
15 lengths of time for the windows):

Time Window 1 | *[Primary Hazard Window]* | **Time Window 2A** | **Time Window 2B**

For example, an earthquake might occur during the primary hazard window, and landslides might directly be triggered in the days to weeks after the earthquake (time window 2A) with an increased probability of occurrence in the months to years after (time window 2B) due to the earthquake's modification of the slope stability. We
20 believe that the manuscript has benefitted from a stronger explanation of this classification. In our revised manuscript **Section 4.1**, we have included additional text where we propose two themes where one can differentiate between one event *triggering* another event, and one event *increasing the probability* of another event. These two themes are (1) *direct vs. indirect sequence of events between the primary and secondary hazard,* and (2) *temporal sequences*, where considering the timing of a sequence of events, and taking both forward
25 looking and retrospective views allows the distinguishing between triggering and increased probability events. Theoretical examples are used to illustrate both themes within the context of the discussion in revised manuscript **Section 4.1**.

**WM Optional Point 1.** As a final suggestion, I would like to see a little bit more emphasis on the probabilistic
30 and quantitative nature of hazard assessment. This is just a suggestion and it is not mandatory, but I do think that quantitative assessment of the hazard (and of the consequent risk) is the only possible strategy to plan rational risk reduction actions.

**Authors**: We agree with this reviewer (and the first reviewer general comments) that there is an important role
35 for quantitative hazard assessment methods. In our revised manuscript we have provided additional discussion of quantitative hazard assessment and interactions likelihoods in our revised manuscript **Section 2.2** (*From Global to Local Multi-Hazard Approaches*), **Section 4.2** (*Triggering Relationships*), and **Section 4.4** (*Catalysing and Impedance Relationships*).

[revised manuscript text omitted]

**Key:**

⇨ *Triggering Relationship*

⬛➤ *Triggering Relationship (forming part of example cascade)*

*Secondary and tertiary hazards may initiate further interactions.*

**Figure 7**. Network of hazard interactions (Example B2) using a hazard/process flow diagram. Using the visualisations constructed in **Figures 2** and **3**, an example of an interaction network (cascade) can be presented. In this example the network is more complex than in Example 1A (**Figure 6**), with three branches and five interaction relationships highlighted here. Three hazard/process groups are included: (**A**) natural hazards, (**B**) anthropogenic processes and (**C**) technological hazards/disasters.

Arrows are used to illustrate interaction relationships, with both triggering relationships (thick block arrows with solid outlines) and relevant catalysing/impedance relationships (thin block arrows with dashed outlines). For clarity of communication, those catalysing/impedance relationships not of relevance to the specific example are not included. See **Figures 2** and **3** caption explanations for further details. Arrows within the example network of hazard interactions are labelled (**1**–**54**) and shaded dark blue to highlight the relevant pathway. This example shows a primary natural hazard triggering (**1**) a primary technological hazard, which in turn triggers (**2**) a primary anthropogenic process. The same primary natural hazard may trigger (**3**) a secondary natural hazard. This secondary natural hazard could then trigger (**4**) a primary technological hazard and (**5**) tertiary natural hazards.

[Figure]

**Figure 8**. Example of vulnerability changes during a network of hazard interactions (cascade). A representation of changing vulnerability during a hazard cascade, where the magnitude of vulnerability is proportional to the size of the box. Following a disaster event, pressures on society, infrastructure and coping capacity are likely to be increased, and thus the vulnerability of a community and its systems/assets to further shocks or hazards may increase.

---

## Author Response (AR2)

**Response to Reviewer Comments (Based on Revised Manuscript)**

**Title: Hazard Interactions and Interaction Networks (Cascades) within Multi-Hazard Methodologies**

**[doi:10.5194/esd-2015-94] submitted to *Earth System Dynamics***

We thank **Reviewer #1 (Reik Donner)** and **Reviewer #2 (Warner Marzocchi)** for their additional corrections and constructive comments on our manuscript, and greatly appreciate the time they have given in reviewing this manuscript. We have addressed all the corrections and comments made by both reviewers in our revised manuscript. Here we present our response to each of the reviewers' comments and explain how we have changed our manuscript to reflect these comments. We also attach a track change document showing changes between this version of the manuscript and the initial one submitted to the journal.

We hope that this final revision is satisfactory, and look forward, if accepted, to dealing with any final copy-editing that might be necessary.

We look forward to your comments.

Joel C. Gill and Bruce D. Malamud
(Department of Geography, King's College London, UK, corresponding author: joel.gill@kcl.ac.uk)

**REVIEWER #1 (COMMENTS TO AUTHOR): Reik Donner [RD]**

**RD General comments:** The authors have carefully revised their manuscript according to the recommendations of both reviewers. I recommend its acceptance in its present form, but suggest considering whether the following points call for corrections in the published version (line numbers refer to the manuscript version without tracked changes):

**Authors:** We are very grateful for the diligence with which you have reviewed this manuscript. We have addressed each of the points that you have listed below. We have again added some clarifying words at the beginning of each of your comments in **[ ]** to give a highlight of what the point is about.

**RD** 1. **[English]** Corrections suggested:
- p.1, l.16: "hazard interaction matrices and hazard/process flow diagrams"
- p.3, l.28: "is turn" => "in turn" (typo)
- p.4, l.127: "dependence of the number"
- p.5, l.15: "recognise that"
- p.5, l.16: "differ from"
- p.9, l.17: "suggest that it is"
- p.9, l.26: remove comma after "whereas"(?)
- p.10, l.13: "i.e., until the end of"
- p.11, ll.26-29: The second sentence appears grammatically incomplete without being explicitly connected with the first one.
- p.13, ll.25-26: "…processes, relevant interactions can be identified…"
- p.14, l.19: "two of these"
- p.14, ll.19-20 mentions two theoretical case studies, while p.16, l.14 refers to three – please be consistent here
- p.16, l.11: "many other" (typo)
- p.16, l.17: "in the spatial and temporal extent"(?)
- p.18, l.3: two times "developed" in one sentence, maybe look for some synonym
- p.19, l.3: "examples… show all…" (typo)
- p.20, l.26: "assumption is made that each hazard"

- p.20, l.3: "may mean that they have"

**Authors**: We thank the reviewer for these helpful observations. We accept the reviewer's suggestions for each of them and have corrected them in the revised manuscript.

**RD** 2. **[Rephrasing]** The first sentences of abstract and introduction, respectively, are rather similar. The authors might think about rephrasing one of them a bit further.

**Authors**: We have modified the first sentence of the introduction so that it is slightly different to the abstract.

**RD** 3. **[Attribution]** (p.10, last paragraph): Maybe briefly relate your corresponding considerations here to the problem of "attribution", which draws an explicit link to ongoing debates especially in the climate change impacts/extreme events community.

**Authors**: We thank the reviewer for this helpful suggestion to enhance the manuscript. We have added a couple of sentences to **Section 4.1** that links this section to the ongoing debates in the climate change/extreme events community. We have introduced two new references here.

- Shepherd, T. G.: A common framework for approaches to extreme event attribution, Current Climate Change Reports, 2(1), 28-38, 2016.
- Stott, P. A., Allen, M., Christidis, N., Dole, R. M., Hoerling, M., Huntingford, C., Pall, P., Perlwitz, J., and Stone, D.: Attribution of weather and climate-related events, in: Climate Science for Serving Society, Asrar, G. R. and Hurrell, J. W. (Eds.), Springer Netherlands, 307–337, 2013.

**RD** 4. **[Networks]** Section 5.4: Conceptually, I understand the term "network" as commonly being used for describing a "substrate" of "connections/interdependencies" between well defined "units" on/between which dynamical processes (may) take place. In this context, I found some formulations like "during a network of hazard interactions" or "networks… propagating" a bit unusual. In my opinion, it is rather the hazards propagating along the network, and "during a network" (temporal perspective) might also be better rephrased as "along a network" (structural perspective) – due to the temporal ordering between primary and secondary hazards, both perspectives are of course equivalent here, but the corresponding sentences just read a bit uncommon to someone working with the concept of networks in different contexts. The authors may wish to reformulate these points or just leave them as they are,

**Authors**: We thank the reviewer for this observation. We have examined **Section 5.4** and rephrased those sentences where the formulation was unusual in the context of "networks".

**REVIEWER #2 (COMMENTS TO AUTHOR): Warner Marzocchi (WM)**

**WM1 [Forecasting Time]:** I am fine with this revision. The authors did a good job in addressing all my comments. Only one minor issue remains that might be corrected. In particular, at page 5 the authors write "Marzocchi et al (2012) proposes that hazard interactions are of greater importance when assessing short term hazard, and less important when considering long term hazard". Actually, this is not what we meant (and I think what we wrote). We think that the need of *modeling interactions* is more necessary in the short term, when, for example, the occurrence of one threat (primary hazard) modifies significantly in the short term the probability to have secondary hazards. In a long term perspective, for instance to calculate the tsunami hazard in the next 50 years, databases already contain information of the fact that most of tsunamis are triggered by earthquakes, and there is no need to make additional calculations to calculate the long term tsunami hazard. But also in the long term it may be *very important* to calculate that earthquakes and tsunami can occur close in time, mostly to evaluate the risk/losses (not the hazard).

**Authors**: We thank the reviewer for this further clarification, which was very helpful in making clear the reviewers perspective on this point. We agree with the reviewer and have amended **Section 2.2** to incorporate these comments and give a more enhanced discussion of Marzocchi *et al.* (2012).

[revised manuscript text omitted]